

# Grounding-line flux formula applied as a flux condition in numerical simulations fails for buttressed Antarctic ice streams

Ronja Reese[1,2], Ricarda Winkelmann[1,2], and G. Hilmar Gudmundsson[3]

[1] Potsdam Institute for Climate Impact Research (PIK), Member of the Leibniz Association, P.O. Box 60 12 03, D-14412 Potsdam, Germany
[2] University of Potsdam, Institute of Physics and Astronomy, Karl-Liebknecht-Str. 24-25, 14476 Potsdam, Germany
[3] Extreme Environments, Northumbria University, Newcastle, UK

*Correspondence to:* Ricarda Winkelmann (ricarda.winkelmann@pik-potsdam.de)

**Abstract.** Currently, several large-scale ice-flow models impose a condition on ice-flux across grounding lines using an analytically motivated parameterization of grounding-line flux. It has been suggested that employing this analytical expression alleviates the need for highly resolved computational domains around grounding lines of marine ice sheets. While the analytical flux formula is expected to be accurate in an unbuttressed flow-line setting, its validity has hitherto not been assessed for

complex and realistic geometries such as those of the Antarctic Ice Sheet. Here the accuracy of this analytical flux formula is tested against an optimized ice flow model that uses a highly-resolved computational mesh around the Antarctic grounding lines. We find that when applied to the Antarctic Ice Sheet the analytical expression provides inaccurate estimates of ice fluxes for almost all grounding lines. Furthermore, in many instances direct application of the analytical formula gives rise to unphysical complexed-valued ice fluxes. We conclude that grounding lines of the Antarctic Ice Sheet are, in general, too highly

buttressed for the analytical parameterization to be of practical value for the calculation of grounding-line fluxes.

## 1  Introduction

Estimating the future impact of the Antarctic Ice Sheet (AIS) on global sea levels invariably involves calculating changes in ice fluxes across grounding lines, as well as determining the migration of the grounding lines themselves. Accurately describing grounding-line dynamics can therefore be considered an essential prerequisite for any numerical ice-flow simulation of marine

ice sheets such as the AIS. Accordingly, over the last decades, considerable efforts have focused on ensuring that large-scale ice-flow models are capable of capturing correctly the dynamical behavior of grounding lines (e.g. Goldberg et al., 2009; Gladstone et al., 2010; Seroussi et al., 2014; Feldmann et al., 2014; Gagliardini et al., 2016; Pattyn et al., 2017). As part of these efforts, several model inter-comparison experiments have been conducted to assess different approaches within the ice-sheet modeling community regarding the numerical modeling of marine-type ice sheets (Pattyn et al., 2012; Drouet et al., 2013; Pattyn et al.,

2013; Asay-Davis et al., 2016). Although still a subject of active research, one of the outcomes of these inter-comparison experiments has been to stress the need for a sufficiently fine resolution of the computational domain around grounding lines. Within the context of the shallow ice-stream computational models — a commonly-used flow approximation for describing the flow of ice streams and ice shelves (e.g., Morland, 1987; MacAyeal, 1989) — it has, for example, been suggested that for



many applications a horizontal resolution of around one ice thickness or less is suitable (Gladstone et al., 2012; Pattyn et al., 2012; Cornford et al., 2016). However, for large-scale ice flow models using uniform grids employing such a high resolution globally for large ice sheets such as the AIS can be computationally prohibitively expensive. As a way of resolving this issue, and to allow for an accurate description of grounding-line dynamics without resorting to high spatial resolution, in a number of

numerical modelling studies a 'flux condition' is imposed at the grounding line whereby the grounding-line flux is prescribed using an analytical expression (e.g., Docquier et al., 2011; Thoma et al., 2014; DeConto and Pollard, 2016; Pattyn, 2017) or grounding line retreat is decided depending on such an expression (e.g. without buttressing parameterisation, Ritz et al., 2015).

The analytical flux expression most often used is based on a theoretical study by Schoof (2007a) and was derived under the assumption that the ice shelf provides *no buttressing* to the ice at the grounding line. The absence of buttressing implies that

the (vertically integrated) horizontal stresses at the grounding line are not affected by the presence of the ice shelf, and were the ice shelf to be removed and replaced by ocean water, the state of stress (in a vertically integrated sense) would remain unaffected (e.g., Schoof, 2007a; MacAyeal and Barcilon, 1988). However, in general, and this is certainly the case for the AIS today (e.g. De Rydt et al., 2015; Fürst et al., 2016; Reese et al., 2017), ice shelves do provide some buttressing. To account for this, numerical models use a modified analytical expression of ice flux based on Schoof (2007a) involving an additional

buttressing parameter ($\theta$) describing the modification in axial stress due to the mechanical impact of the ice shelf on the stress state at the grounding line. The buttressing parameter ($\theta$) needs to be calculated by the numerical ice flow model, and then inserted into the analytical flux expression. The resulting flux is then used by the corresponding numerical model as a flux condition along all grounding lines.

Previous numerical model inter-comparison experiments (Pattyn et al., 2012) have shown that in the unbuttressed case there

is, in general, a good agreement between the analytically and numerically calculated ice fluxes for steady-state conditions. For one particular synthetic model setup, Gudmundsson (2013) also found a good agreement between analytically and numerically calculated ice fluxes for buttressed ice. The question now arises as to how accurately the analytical expression predicts grounding-line ice fluxes for *realistic* geometries such as that of the present-day AIS. More specifically, if one were to apply sufficiently high resolution around all Antarctic grounding lines, would fluxes calculated directly by such a high-resolution

numerical model agree with the predictions of the analytical flux formula? Answering this question is the subject of this study.

Here we assess the accuracy of the analytical flux formula for calculating ice fluxes across grounding lines of present-day Antarctica. We do this by comparing predicted analytical fluxes with independently numerically calculated ice fluxes using the ice-flow model Úa (Gudmundsson, 2013). The ice flow model is applied continent-wide, using high spatial resolution around all grounding lines of few hundreds of meters.

The paper is structured as follows: First, we give a brief overview over the flux formula derived by Schoof (2007a), and discuss several different approaches to quantifying ice-shelf buttressing. We then describe our numerical ice flow model Úa, and the model initialization procedure in Sect. 2. The following Sect. 3 on the comparison between numerically calculated grounding-line ice fluxes and those by the flux formula forms the main part of the paper. This is followed by a discussion of the results and final conclusions, Sect. 4 and Sect. 5.



## 2 Ice shelf buttressing and grounding-line ice flux

In Schoof (2007a), an expression for the grounding-line flux ($q$) of marine ice sheets is derived. While the analysis is primarily focused on a flow-line configuration where ice-shelf buttressing plays no role, Schoof (2007a) also estimates how the flux might be affected by a reduction $\theta$ in axial stress at the grounding line due to ice-shelf buttressing. The resulting analytical flux expression is

$$q(x) = \theta^{\frac{nm}{m+1}} \rho_i h^{\frac{1+m(n+3)}{m+1}} \left( \frac{1}{4^n} A(\rho_i g)^{n+1} (1 - \rho_i/\rho_w)^n C^{1/m} \right)^{\frac{m}{m+1}} \tag{1}$$

where $q$ is the ice flux across the grounding line, $h$ the ice thickness, $\rho_i$ the ice density, $\rho_w$ the density of ocean water and $g$ the gravitational acceleration (please note that in the related Eq. 17 of Gudmundsson (2013) for the flux $q$ there is a typo in the exponent of the basal slipperiness $C$). For grounded ice, the tangential component of the basal traction ($\tau_b$) is related to the basal velocity ($v_b$) through the Weertman-type sliding law

$$\tau_b = C^{-1/m} |v_b|^{1/m-1} v_b, \tag{2}$$

where $C$ is the basal slipperiness, and $m$ the stress exponent, while deviatoric stresses and strain rates $\dot{\epsilon}_{ij}$ in ice flow are linked via Glen's flow law

$$\dot{\epsilon}_{ij} = A\tau^{n-1}\tau_{ij}, \tag{3}$$

with $\tau = \sqrt{\tau_{ij}\tau_{ij}/2}$ the second invariant of the deviatoric stress tensor, exponent $n$ (often set to 3) and rate factor $A$. Here $\tau_{ij}$ denote the components of the deviatoric stress tensor and $\dot{\epsilon}_{ij}$ the components of the strain rate tensor.

As mentioned above $\theta$ is a scalar quantity that describes the deviation in deviatoric axial stress at the grounding line from the unbuttressed situation. For an unbuttressed grounding line in one horizontal dimension (i.e. no variations in any quantities transverse to the flow direction) and assuming that the $x$-axis of the coordinate system is aligned with the flow, we have $\tau_{xx} = \tau_f$ (see Appendix A) where

$$\tau_f = \frac{\rho_i g}{4} \left( 1 - \frac{\rho_i}{\rho_w} \right) h. \tag{4}$$

In the buttressed case, $\tau_{xx}$ is however no longer necessarily equal to $\tau_f$, and $\theta$ is defined as

$$\theta^{1HD} = \frac{\tau_{xx}^{1HD}}{\tau_f}. \tag{5}$$

We have here used the superscript $1HD$ to indicate that this definition of $\theta$ is only unambiguous in the one horizontal dimensional situation (1HD). In the more general two horizontal dimensional situation (2HD), where the flow direction is not necessarily aligned with the (horizontal) normal to the grounding line, several different definitions of $\theta$ are possible, and in the literature at least three different definitions of $\theta$ have been suggested. In the following we denote these by $\theta_1$, $\theta_2$, and $\theta_3$, with

$$\theta_1 = \frac{\boldsymbol{n}_1 \cdot \boldsymbol{R} \boldsymbol{n}_1}{2\tau_f}, \tag{6}$$





where $\boldsymbol{n}_1$ is a normal to the grounding line pointing horizontally outwards from the grounded ice into the ice shelf, and

$$\theta_2 = \frac{\boldsymbol{n}_1 \cdot \boldsymbol{\tau} \boldsymbol{n}_1}{\tau_f}, \tag{7}$$

and

$$\theta_3 = \frac{\boldsymbol{n}_2 \cdot \boldsymbol{\tau} \boldsymbol{n}_2}{\tau_f}, \tag{8}$$

where $\boldsymbol{n}_2$ is the direction of ice flow at the grounding line and

$$\boldsymbol{\tau} = \begin{pmatrix} \tau_{xx} & \tau_{xy} \\ \tau_{xy} & \tau_{yy} \end{pmatrix}, \tag{9}$$

is the (horizontal) deviatoric stress tensor, and

$$\boldsymbol{R} = \begin{pmatrix} 2\tau_{xx} + \tau_{yy} & \tau_{xy} \\ \tau_{xy} & \tau_{xx} + 2\tau_{yy} \end{pmatrix}, \tag{10}$$

the tensor of resistive stresses. In the 1HD unbuttressed case where $\boldsymbol{n}_1 = \boldsymbol{n}_2$, $\tau_{xx} = \rho_i g h (1 - \rho_i/\rho_w)/4$, and $\tau_{yy} = \tau_{xy} = 0$, all these three definitions of $\theta$ result in $\theta_1 = \theta_2 = \theta_3 = 1$. The first definition (i.e. $\theta_1$) has, for example been used by Gudmundsson (2013) to diagnose buttressing at the grounding line of an idealized setup, the second definition by Pollard and DeConto (2012), Thoma et al. (2014) and Pattyn (2017) as a flux condition, and the third one by Fürst et al. (2016) to diagnose 'flow-buttressing' within Antarctic ice shelves. Note however that for instance Pollard and DeConto (2012, see section 2.3), Thoma et al. (2014, see section 4.4), Fürst et al. (2016, see Supplementary Eq. 2) and Pattyn (2017, see Eq. 20) appear to use a different expression for $\tau_f$, with $\tau_f = \rho_i g h (1 - \rho_i/\rho_w)/2$, in which case $\theta = 1/2$ in the unbuttressed case and $\theta$ in Eq. (1) must be replaced by $2\theta$.

The definition of $\theta_1$ is motivated by the form of the boundary condition at the calving front in the shallow ice-stream approximation (see Appendix A). For $\theta_1 = 1$ the normal traction at the grounding line equals that of a calving front. In the general 2HD situation, this same interpretation does not hold for the definitions of $\theta_2$ and $\theta_3$. If $\theta_1 > 1$ the ice shelf can be considered to be 'pulling' the ice at the grounding line, while $\theta_1 < 1$ implies that the ice shelf is reducing the normal traction at the grounding line. Note that for all these three different definitions, it is possible for $\theta$ to become negative. If, however, a negative $\theta$ value is inserted into Eq. (1), the resulting value for the flux $q$ is a negative or even a complex number for most combinations of $n$ and $m$ — a clear indication that the analytical flux formula fails in such situations. Only the specific combinations of $n$ and $m$ such that $nm/(m+1) = 2k$ for $k \in \mathbb{N}$ (for instance the combination $n = 3$ and $m = 2$) 'fix' the flux back to a positive real number, however they introduce a non-substantiated dependency between the flow law and the sliding law. Furthermore, for these combinations and $\theta < 0$, enhanced buttressing - inconsistently - yields an increase in ice flux. Physically, $\theta_1 < 0$ corresponds to a situation where the traction vector at the grounding line points in upstream direction. One possible situation giving rise to $\theta_1 < 0$ would be $\tau_{xx} < 0$ while $\tau_{yy} = 0$, with $x$ being the flow direction and the grounding line aligned with the $y$ axis. In this case, the ice at the grounding line experiences compression in along-flow direction and, hence, longitudinal strain rates are negative and ice velocities become smaller as the grounding line is approached from upstream direction. Another situation giving rise to $\theta_1 < 0$ is that of equal transversal compression and vertical extension of the ice column at the grounding line, i.e. $\tau_{yy} = -\tau_{zz} < 0$ while $\tau_{xx} = 0$.





## 2.1 Model description

We diagnose the fluxes at the grounding line with the finite-element ice-flow model Úa (Gudmundsson, 2013). The flow model Úa has been used to calculate the ice-flow for various geometries involving ice-shelf buttressing (e.g. De Rydt and Gudmundsson, 2016; Royston and Gudmundsson, 2016; Gudmundsson et al., 2017), and results obtained by the model submitted to a number of model inter-comparison experiments (MISMIP, Pattyn et al., 2012) and (MISMIP3d, Pattyn et al., 2013). The model employs an unstructured grid and hence allows for resolving the grounding line zone locally with high resolution. The model further allows for nodal-based or element-based, and simultaneous inversion of the ice rate factor $A$ and the basal slipperiness $C$ using either Bayesian or Tikhonov type regularization.

Here we use Úa to solve the shallow ice-stream equations (e.g., Morland, 1987; MacAyeal, 1989) in a diagnostic mode using a Weertman-type sliding law (see Eq. 2) and Glen's flow law (see Eq. 3). In the glaciological literature the shallow ice-stream equations are also referred to as the Shallow-Shelf/Shelfy-Stream Approximation and often abbreviated as SSA. In 2HD the SSA momentum equations are

$$\nabla_{xy} \cdot (h\,\boldsymbol{R}) - \boldsymbol{\tau}_{bh} = \rho_i g h \nabla_{xy}\, s + \frac{1}{2} g h^2 \nabla_{xy}\, \rho_i, \tag{11}$$

where

$$\nabla_{xy} = (\partial_x, \partial_y) \tag{12}$$

and $\boldsymbol{R}$ is the tensor of resistive stresses given by Eq. (10), $h$ is the ice thickness, $s$ the ice surface elevation, $\rho_i$ the vertically averaged density, and $\boldsymbol{\tau}_{bh}$ is the horizontal part of the bed-tangential basal traction $\boldsymbol{\tau}_b$. Where the ice is floating $\boldsymbol{\tau}_{bh} = 0$. In the SSA the flotation criterion has the form $h < h_f$ with

$$h_f = (S - B)\rho_w/\rho_i, \tag{13}$$

where $S$ is the ocean surface, $B$ the bedrock, and $\rho_w$ is the ocean density. The flotation criterion in Úa is evaluated at each integration point of the elements of the finite element mesh and the basal drag term evaluated accordingly through a standard finite-element procedure involving element-wise integration.

## 2.2 Methodology

Using the ice flow model Úa, we calculate ice velocities for the entire Antarctic Ice Sheet, including all ice shelves. The SSA equations are solved throughout the computational domain. Stress boundary conditions (i.e. Neumann boundary conditions) are applied at the margins of the computational domain. Since the modeling domain covers the whole of the AIS, no inflow or outflow boundary conditions (i.e. Dirichlet boundary conditions) need to be applied at any sections of the boundary.

Two different computational meshes were generated and the sensitivity of the results evaluated using linear (3-node), quadratic (6-node), and cubic (10-node) triangular elements. All results presented here were obtained using a very high resolution mesh generated with the finite-element mesh generator *Gmsh* (Geuzaine and Remacle, 2009) with $1,360,894$ triangular



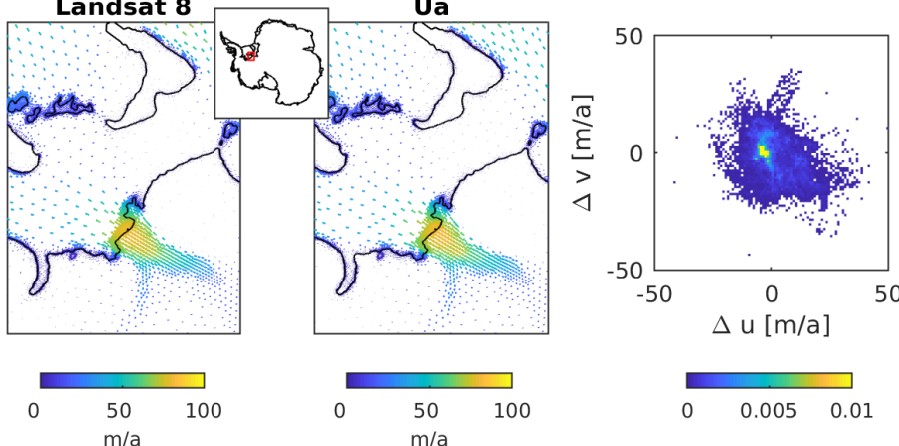

**Figure 1.** Observed (left panel) and modelled (middle panel) ice velocities in the region of Institute Ice Stream. The inset displays the location of the plotted area in Antarctica. Grounding lines are shown as black lines and colorscales indicate the speed. The right panel shows a normalised bivariate histogram of the velocity residuals which are the differences between modeled and observed velocities within this area, that is, $\Delta u = u_{\mathrm{modeled}} - u_{\mathrm{observed}}$ and $\Delta v = v_{\mathrm{modeled}} - v_{\mathrm{observed}}$, and $u$ and $v$ are the horizontal components of the surface velocity vector, respectively.

linear elements and $689,042$ nodes. Within $5\,\mathrm{km}$ distance to the grounding line, the mesh was refined such that element sizes decrease towards the grounding line to a maximum size of $250\,\mathrm{m}$ directly at the grounding line. Overall, the elements have a maximal size of $179,307\,\mathrm{m}$ in the interior of the continent and minimal size of $56\,\mathrm{m}$ along the grounding line. Mean element size is $1596\,\mathrm{m}$ and median $480\,\mathrm{m}$. A regional example of the mesh is given in Fig. S.1. The robustness of the results was also

tested based on the mesh used in Reese et al. (2017), as discussed in Appendix B.

    Ice thickness and bed geometry input is based on the Bedmap2 estimates (Fretwell et al., 2013). Vertically averaged ice densities were calculated using firn thickness fields from RACMO2 (Lenaerts et al., 2012) and assuming a constant ice density of $910\,\mathrm{kg\,m^{-3}}$ and a firn density of $500\,\mathrm{kg\,m^{-3}}$. Resulting densities range from $770\,\mathrm{kg\,m^{-3}}$ to $910\,\mathrm{kg\,m^{-3}}$ and the horizontal gradients in vertically averaged densities are hence small, see Fig. S.2. In a few places the bathymetry around grounding lines

was vertically modified to improve its alignment with Bindschadler et al. (2011), with vertical adjustments of maximally $50\,\mathrm{m}$ being allowed.

    For the entire Antarctic setup we inverted for basal slipperiness $C$ (see Eq. 2) and ice softness fields $A$ (see Eq. 3) to match observed 2015/2016 velocities derived from Landsat 8 imagery (Gardner et al., 2017). The stress exponent of Glen's flow law was set to $n = 3$ and we repeated the inversion for a whole sequence of sliding law exponents $m = 1, 2, 3, 4, 5, 7, 9, 11$. We

inverted for $A$ and $C$ over the computational nodes using Tikhonov type regularization. The inversion procedure minimizes the function

$$J(u, p) = I(u) + R(p)$$



with respect to $p$, where $p$ stands for model parameters to be determined (i.e. $A$ and $C$, here $C$ was set to $0$ within ice shelves), $u$ are modeled surface velocities, $I$ the data misfit function, and $R$ the regularization term. The misfit function $I$ has the form

$$I(f) = \frac{1}{2\mathcal{A}} \int (\boldsymbol{v}_{modeled} - \boldsymbol{v}_{observed})^2 / e^2 \, dA \tag{14}$$

where $\mathcal{A} = \int dA$ is the total area, $\boldsymbol{v}_{modeled}$ and $\boldsymbol{v}_{observed}$ modeled and observed velocities, respectively, and $e$ data errors. The regularization function $R$ has the form

$$R = \frac{1}{2\mathcal{A}} \int \left( \gamma_s^2 \left( \nabla \left( \log_{10}(p) - \log_{10}(\hat{p}) \right) \right)^2 + \gamma_a^2 \left( \log_{10}(p) - \log_{10}(\hat{p}) \right)^2 \right) dA$$
$$= \frac{1}{2\mathcal{A}} \int \left( \gamma_s^2 \left( \nabla \log_{10}(p/\hat{p}) \right)^2 + \gamma_a^2 \left( \log_{10}(p/\hat{p}) \right)^2 \right) dA \tag{15}$$

where $\gamma_a$ and $\gamma_s$ are regularization parameters, and $\hat{p}$ the *a priori* values for model parameters. Inversions were done for a wide ranges of $\gamma_s$ and $\gamma_a$ and optimal values determined from an $L$-curve analysis. In the results shown here, we use $\gamma_a = 1$ and $\gamma_s = 10,000\,\mathrm{m}$. However our results are insensitive to the particular values chosen.

For $\gamma_s = 10,000\,\mathrm{m}$, $\gamma_a = 1$ and the sliding exponent $m = 3$, the corresponding basal slipperiness $C$ and the ice rate factor $A$ distributions are shown in Figs. S.3 and S.4. The average difference between modeled and observed ice speed is $29$ meters per year with a median of $13$ meters per year and a root mean square error of $103$ meters per year. The measured and modeled velocity fields for the region of Institute Ice Stream are displayed in the left and middle panels of Fig. 1. They agree well in this area, as the residual histogram for this region shows in the right panel, but also for the entire continent, see Fig. S.5. As a consequence of our inverse methodology, modeled ice velocities are in close accordance with measurements.

From the modeled stresses obtained with our ice-flow model we calculate the buttressing parameter $\theta$ as defined in Sect. 2. We do this for each of the three different definitions for $\theta$ (see Eqs. 6, 7, and 8). We then calculate the *analytical* fluxes predicted by the flux formula, i.e. Eq. (1). Note that we refer to these fluxes as 'analytical' fluxes although their calculation involves the use of our numerical ice-flow model for estimating the buttressing number $\theta$.

We also calculate *modeled* grounding-line fluxes from modeled ice velocities. Since our modeled velocities are in a good agreement with observed velocities, these modeled grounding-flux estimates will be in an equally good agreement with fluxes estimated directly from observed velocities. The analytical and the modeled flux estimates are then compared and analysed.

When calculating grounding-line fluxes we interpolate nodal quantities of the computational mesh onto the (calculated) grounding line. The grounding line does not, as such, enter the numerical calculations done by our numerical ice flow model. As described in Sect. 2.1, it is the flotation mask — evaluated at the integration points — that determines the impact of the basal drag term. However, in a post-processing step we determine the positions of the grounding lines from the flotation mask. Our approximation of the grounding line is a piecewise linear curve, with each linear segment representing the grounding line within a given computational element (see Figs. S.1 and B1). We then interpolate nodal values onto the central point of each such linear segment. This same procedure is employed when calculating both analytical and modeled fluxes.





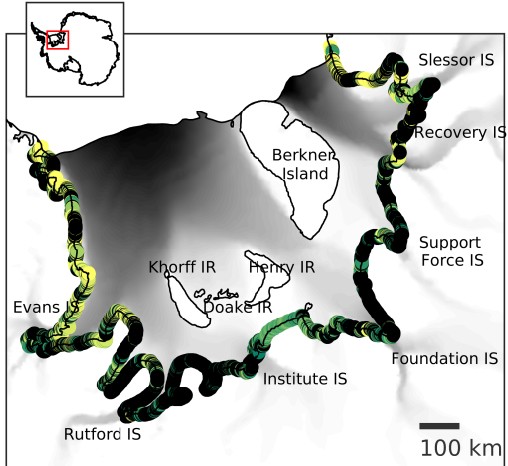
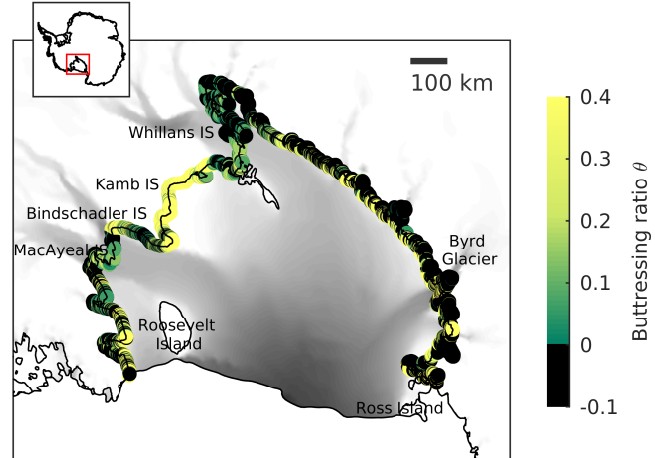

**Figure 2.** Buttressing ratio $\theta_1$ along the grounding lines of Filchner-Ronne Ice Shelf (left panel) and Ross Ice Shelf (right panel). Insets indicate the ice shelves' locations in Antarctica, correspondingly. Regions where the grounding line is 'over-buttressed', that is, $\theta \leq 0$, are displayed in black. Modelled speed is plotted in gray ranging up to $1,500 \text{ ma}^{-1}$. Grounding line and ice front locations are indicated in black. IS denotes ice streams, IR denotes ice rises or rumples.

## 3 Results

From the numerically modeled stress field we calculate the buttressing parameter $\theta_1$ (given by Eq. 6) for all grounding lines of the Antarctic setup described in Sect. 2.2. While we here focus on the buttressing parameter $\theta_1$, our findings are independent of the exact definition of $\theta$, the choice of the sliding law exponents $m$, the mesh and the details of the inverse methodology
applied (see Appendix B).

We find that the grounding lines of Filchner-Ronne and Ross ice shelves are, in general, highly buttressed with buttressing values significantly different from unity (see Fig. 2). Typically, $\theta_1 \leq 0.4$, and in many cases $\theta_1 < 0$. Among the ice streams of these two biggest ice shelves of the AIS, the dormant Kamb Ice Stream is the relatively least buttressed one, with $\theta_1 \approx 0.4$. Over all other ice streams $\theta$ values are even smaller. Negative $\theta$ values are also found over grounding-line segments located
between active ice streams, for example along the grounding line running between the Rutford and Institute Ice Streams.

An example of an ice stream where $\theta_1 < 0$ over most of its grounding line is the Institute Ice Stream (see Figs. 3 and 4). Inspection of the velocity field in the vicinity of the grounding line of that ice stream reveals that ice flow velocities decrease with distance as the grounding line is approached from up-stream direction. Consequently, both along-flow strain rates and along-flow deviatoric stresses are negative (compressive). This general feature of ice flow around the grounding line of Institute
Ice Stream implies that its grounding line is 'over-buttressed' with the traction vector at the grounding line pointing in inland direction. Hence, independently of our numerical simulation of the stress field, it is clear that for this ice stream $\theta$ must be negative.





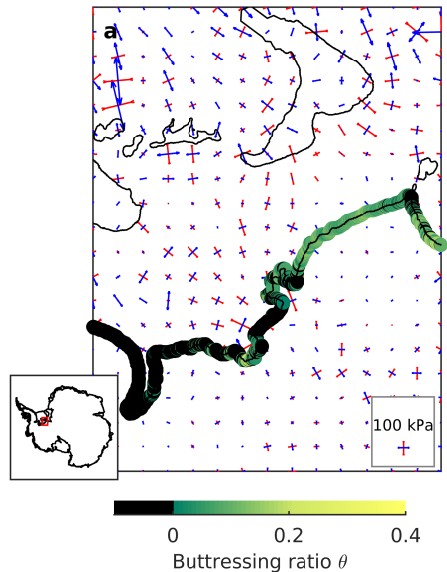
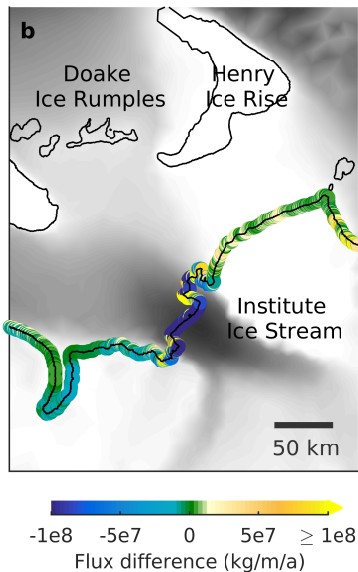

**Figure 3.** Buttressing ratio and differences in grounding line flux for Institute Ice Stream draining into the Filchner-Ronne Ice Shelf (location shown in inset). (a) Buttressing values $\theta_1$ are displayed along the grounding line and principle deviatoric stresses are shown, with compression in red and extension in blue. The length of the vectors indicate the magnitude of each principle stress. (b) Differences between analytical and modeled fluxes and observed ice velocities ranging up to $500 \, \mathrm{m \, a^{-1}}$ (Gardner et al., 2017). Analytical fluxes are set to 0 where $\theta_1 < 0$. Grounding line positions are indicated in black.

As discussed in Sect. 2 the analytical flux formula (Eq. 1) is clearly not applicable in situations where $\theta$ becomes negative (see also Sec. 2). As $\theta$ is found to become negative over large sections of the grounding lines of many the ice streams of the two largest Antarctic ice shelves, i.e. Ross and Filchner-Ronne ice shelves, it follows that the formula can *not* be used to calculate grounding-line ice fluxes over significant parts of the AIS.

We furthermore compare analytical and numerically modeled grounding-line ice fluxes in all regions where $\theta_1 \geq 0$, i.e. where the application of the analytical flux formula (Eq. 1) results in real-valued ice fluxes. In particular we compare both the flux values point-wise along all grounding lines (Fig. 5) and the total cumulative fluxes over grounding-lines of ice streams and ice shelves (Table 1). When comparing cumulative analytical fluxes, we are forced to assume values for those sections of grounding lines for which $\theta$ is negative (and $q$ complex). There we assume $q = 0$, which is equivalent to setting $\theta = 0$.

In general, we find significant differences between analytically calculated and numerically modeled flux values. Analytical fluxes are much lower than modeled in many locations of Filchner-Ronne Ice Shelf, especially along the grounding lines of the Rutford, Institute and Moeller ice streams (Fig. 5). However, cumulative analytical fluxes over all grounding lines of the Filchner-Ronne Ice Shelf are about 30% larger than modeled for $\theta_1$, and this difference is considerably larger for $\theta_2$ and $\theta_3$ (Table 1). Similar disagreement between analytical and modeled fluxes is found for the Siple Coast Ice Streams such as

Bindschadler and MacAyeal Ice Streams, and for Byrd Glacier (Panel b of Fig. 5). For Ross Ice Shelf the overall difference is



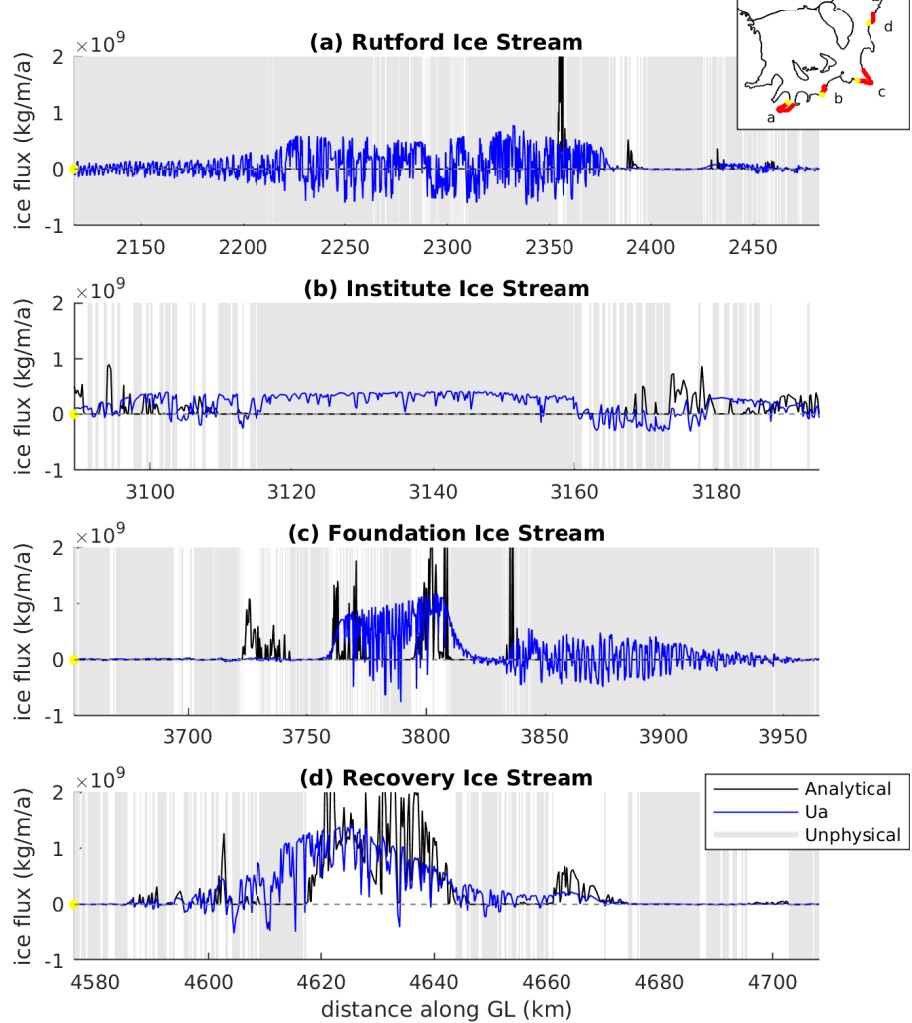

**Figure 4.** Comparison of fluxes calculated with Úa (blue) and analytical fluxes (black) along the grounding lines of four major ice streams draining into Filchner-Ronne Ice Shelf. Locations where the flux formula provides unphysical results are marked in grey. Plotted grounding line segments are located as displayed in the inset with western margins indicated by a yellow dot.

only 5%, but given the fact that $\theta_1$ is negative over significant sections of its grounding line (where we set the analytical flux values to zero), this agreement appears somewhat fortuitous.

For other ice shelves, cumulative fluxes are generally underestimated by the flux formula. Analytical fluxes for Pine Island Glacier and Thwaites Glacier, for example, deviate by $-33\%$ and $-52\%$ from the modeled fluxes, respectively. For George VI ice shelf, cumulative analytical fluxes are several times smaller than modeled ones (Table 1).



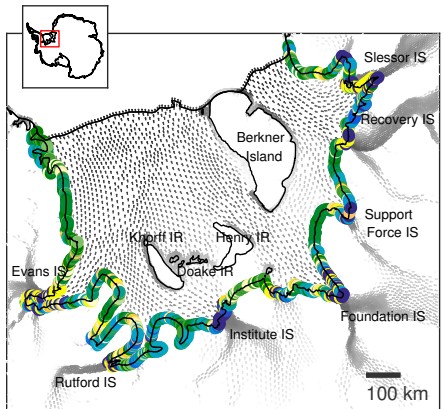
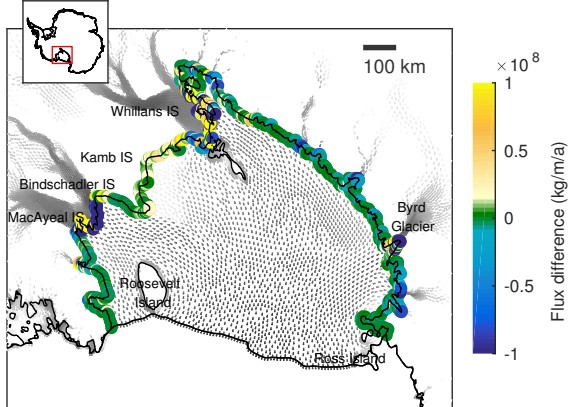

**Figure 5.** Difference between the analytical and the modeled fluxes along the grounding lines of Filchner-Ronne Ice Shelf (left panel) and Ross Ice Shelf (right panel). Analytical fluxes are calculated based on $\theta_1$ defined in Eq. 6. In locations where the formula yields unphysical results, fluxes are set to zero. Grey arrows show the modeled ice flow. IS denotes ice streams, IR denotes ice rises or rumples. Grounding line and ice front locations are indicated in black.

The analytical flux formula tends to strongly overestimate fluxes over grounding lines where ice flow is approximately tangential to the grounding line. This failure of the flux formula to correctly predict fluxes in such circumstances is not surprising as the underlying assumptions of the formula are clearly not met in such situations. Nevertheless, this demonstrates the inherent conceptual difficulties in applying the formula to the Antarctic Ice Sheet.

Moreover, the analytical formula produces much higher spatial variability in fluxes than the numerically modeled ones. This can be clearly seen in Fig. 4 where analytical and modeled ice fluxes are plotted along the grounding lines of Rutford, Institute, Foundation and Recovery ice streams. Here, gray background indicates sections of the respective grounding lines where the flux formula yields unphysical results. Variability in fluxes calculated with Úa occurs when ice flow is nearly aligned with the grounding line.

We test the sensitivity of our analytical flux calculations to different degrees of regularization ($\gamma_s$ and $\gamma_a$) and different values of the sliding law stress exponent ($m$), for which our findings are summarized in Figs. B3 and B5. Numerically modeled fluxes are, as expected, mostly independent of the value of the sliding law stress exponent $m$. This can be considered to be a consequence of the inversion procedure, which ensures that modeled velocity fields agree closely with measured data, independently of the value of $m$. On the other hand, analytically calculated flux values are highly sensitive to the value of

$m$ (see Fig. B3). For example, cumulative analytical fluxes for Filchner-Ronne Ice Shelf increase by about a factor of five as $m$ is changed from 1 to 7, while numerically modeled fluxes change by less than 10 %. Numerically modeled fluxes are also insensitive to the exact degree of regularization applied, whereas analytically calculated flux values change significantly (Fig. B5). The dependency of the analytically calculated fluxes on the amount of regularization used in the numerical model is due to the impact regularization has on modeled stresses and, therefore, on the value of $\theta$.



**Table 1.** Ice flux integrated along the grounding lines of Antarctic ice shelves. $Q_{\text{Úa}}$ denotes the modeled ice flux with Úa, $Q_1$ was derived from the analytical flux formula based on $\theta_1$, $Q_2$ based on $\theta_2$ and $Q_3$ based on $\theta_3$, respectively. Last column shows the deviation of the analytical flux $Q_1$ from the modeled $Q_{\text{Úa}}$.

| Ice shelf | $Q_{\text{Úa}}$ [Gt a$^{-1}$] | $Q_1$ [Gt a$^{-1}$] | $Q_2$ [Gt a$^{-1}$] | $Q_3$ [Gt a$^{-1}$] | $(Q_1 - Q_{\text{Úa}})/Q_{\text{Úa}}$ [%] |
|---|---|---|---|---|---|
| Filchner-Ronne | 216 | 282 | 694 | 755 | 30 |
| Pine Island | 123 | 82 | 148 | 190 | -33 |
| Ross | 120 | 126 | 280 | 155 | 5 |
| Thwaites | 117 | 57 | 82 | 133 | -52 |
| Getz | 91 | 27 | 52 | 60 | -70 |
| Totten | 65 | 44 | 158 | 243 | -32 |
| George VI | 64 | 9 | 21 | 21 | -85 |
| Amery | 55 | 16 | 135 | 56 | -70 |
| Moscow-University | 43 | 16 | 44 | 120 | -63 |
| West | 40 | 27 | 33 | 49 | -32 |
| Shackleton | 37 | 20 | 62 | 61 | -48 |
| Crosson | 34 | 17 | 38 | 38 | -51 |
| Larsen C, D | 25 | 9 | 19 | 38 | -64 |
| Brunt/Stancomb-Wills | 22 | 18 | 24 | 40 | -16 |
| Fimbul | 21 | 7 | 15 | 15 | -67 |
| Stange | 16 | 3 | 13 | 15 | -81 |
| Riiser-Larsen | 12 | 9 | 20 | 25 | -29 |
| Dotson | 11 | 2 | 13 | 19 | -84 |

We also compare analytical fluxes as calculated using the three different definitions (6), (7) and (8) for $\theta$. While overall spatial variability of $\theta$ is similar for these three definitions, with all definitions giving rise to extended areas of negative $\theta$ values, the cumulative flux for the alternative definitions $\theta_2$ and $\theta_3$ are generally higher than for $\theta_1$ (see also Fig. B4).



## 4  Discussion

The analytical grounding-line flux formula Eq. (1) was derived for a flow-line configuration (Schoof, 2007a), and there is no
reason to doubt its validity in that particular case. When applied to a flow-line configuration, many current ice-flow mod-
els employing the shallow ice-stream approximation (SSA) with Weertman-type sliding law, have demonstrated an excellent

agreement between modeled and analytical grounding-line fluxes (Pattyn et al., 2012). Ice fluxes and grounding line positions
calculated with the ice flow model Úa also agree closely with those predicted by Eq. (1) where such an agreement is to be
expected. The inclusion of the buttressing parameter $\theta$ was used by Schoof (2007a) to illustrate the potential impacts of ice-
shelf buttressing on ice flux, provided its effects were sufficiently small as not to invalidate too strongly the basic assumption
of a flow-line setting. However, we find that most of the grounding lines of the AIS are highly buttressed with $\theta$ significantly

different from unity. Therefore the assumptions of the flux formula are in most cases not met. When applied to the current ge-
ometry and the current flow field of the AIS, the flux formula predicts either unphysical or highly inaccurate flux values when
compared to modeled ones. While we have done the comparison with numerically modeled fluxes, comparison with observed
fluxes — as calculated from measured surface velocities, observed grounding-line positions, and measured ice thicknesses
— would not alter our conclusions as, due to our inversion procedure, observed and modeled surface velocities are in good

agreement.

The strongest indication that the analytical flux formula fails when applied to the Antarctic Ice Sheet is arguably the fact
that it predicts non-real valued fluxes over significant parts of Antarctic grounding lines. This happens whenever $\theta$ becomes
negative except for specific combinations of $n$ and $m$ which also yield unphysical solutions as discussed in Sect. 2. As we
point out above, even a cursory inspection of the velocity field of the AIS suffices to show that $\theta$ is negative for a number of

grounding lines (e.g. the Institute Ice Stream grounding line). Hence, the occurrence of negative $\theta$ values is not simply a feature
of our particular numerical approach, but a general aspect of the current ice-flow regime of the AIS.

As analytical ice fluxes are strongly dependent on ice thickness ($h$) at the grounding line, they depend somewhat on the
specifications of the numerical model: the exact location of the grounding line is influenced by the mesh resolution used by the
model. The resulting error is an example of a discretization error that becomes smaller as the mesh is refined. Other numerical

models using a different computational mesh may locate the grounding line differently and hence calculate flux values different
to some extent. We tested the dependency of our modeled ice fluxes to grid resolution by using several different meshes — an
example of two such meshes is given in Fig. B1 — and found none of our main conclusions to be affected by differences in
mesh resolution.

As measured by the buttressing parameter $\theta_1$, almost all grounding lines of the AIS can be considered to be strongly but-

tressed with, in most cases, $\theta < 0.4$. Hence, theoretical concepts based on the assumption of none, or insignificant, ice-shelf
buttressing may not apply to present-day Antarctica. One such theoretical prediction of considerable relevance for the possible
future of the AIS relates to the stability of its grounding lines. In the absence of ice-shelf buttressing, grounding-line stability is
predicted to be related to local bed slope (Weertman, 1974; Thomas and Bentley, 1978; Schoof, 2007a, b, 2011). However, in
the presence of ice-shelf buttressing no such simple conclusions can be drawn (e.g Goldberg et al., 2009; Gudmundsson et al.,



2012; Gudmundsson, 2013; Pegler, 2016). Possibly, rather than being dominated by local bed slope, the stability regime of the Antarctic Ice Sheet is to a leading-order dependent on the properties of the ice shelves downstream of its grounding lines (e.g. geometry and structural integrity). Further work is needed to address the question of the stability of Antarctica's grounding lines.

## 5 Conclusions

In our study, we compare grounding-line ice fluxes obtained by an ice-sheet model with fluxes predicted by an analytical flux formula based on Schoof (2007a). The formula includes a parameter ($\theta$) to account for ice-shelf buttressing, and the resulting flux is sometimes applied as a grounding-line flux condition in numerical simulations. We find that the formula results in unphysical and grossly inaccurate grounding-line fluxes for most of the AIS. We furthermore find that almost all Antarctic grounding lines are highly buttressed, suggesting that the assumption of the analytical flux formula of weakly buttressed grounding lines is not met for the current configuration of the Antarctic Ice Sheet.





**Appendix A: Vertically integrated stress boundary condition at a free calving front**

A derivation of the boundary condition at the calving front for the momentum equations in 2HD can be found for example in Cuffey and Paterson (2010) and van der Veen (1999). At the calving face:

$$\int_b^s \sigma \cdot n_{\mathrm{cf}}\, dz = -\int_b^S p_w \cdot n_{\mathrm{cf}}\, dz$$

where $n = (n_x, n_y, 0)$ is the normal of the calving front pointing outwards, $s$ the ice surface, $S$ the sea-level and $p_w$ is hydrostatic pressure in the ocean $p_w = \rho_w g(S - z)$. The balance in $x$-direction reads:

$$\int_b^s (\sigma_{xx} n_x + \sigma_{xy} n_y)\, dz = \int_b^S -\rho_w g(S - z) n_x\, dz = -\frac{\rho_i^2 g}{2\rho_w} h^2 n_x \tag{A1}$$

We can rewrite $\sigma_{xx} = 2\tau_{xx} + \tau_{yy} + \sigma_{zz}$ (since $\sigma_{xx} = \tau_{xx} + p$, $\sigma_{zz} = \tau_{zz} + p$ and $\tau_{xx} + \tau_{yy} = -\tau_{zz}$). Under the assumptions of the hydrostatic approximation, $\sigma_{zz} = -\rho_i g(s - z)$. The vertically integrated horizontal stress balance equals

$$\int_b^s (\sigma_{xx} n_x + \sigma_{xy} n_y)\, dz = 2h\tau_{xx} n_x + h\tau_{yy} n_x + h\tau_{xy} n_y - \frac{\rho_i g}{2} h^2 n_x, \tag{A2}$$

since $\tau_{xx}, \tau_{yy}, n_x$ and $n_y$ do not vary vertically. Inserting this in Eq. A1 yields:

$$(2\tau_{xx} + \tau_{yy}) n_x + \tau_{xy} n_y = \frac{\rho_i g}{2}\left(1 - \frac{\rho_i}{\rho_w}\right) h n_x. \tag{A3}$$

Similarly for the $y$-direction. This can be abbreviated as

$$R \cdot n = \frac{\rho_i g}{2}\left(1 - \frac{\rho_i}{\rho_w}\right) h n. \tag{A4}$$

Following Gudmundsson (2013) we obtain the normal buttressing value which compares the RHS and LHS of the equation above in direction of the normal $n$ at the grounding line:

$$\theta = \frac{n \cdot Rn}{\frac{\rho_i g}{2}\left(1 - \frac{\rho_i}{\rho_w}\right) h} = \frac{n \cdot Rn}{2\tau_f}. \tag{A5}$$

In the case of a laterally uniform unconfined ice shelf with $\tau_{yy} = 0$ and $\tau_{xy} = 0$, this reduces to $\tau_{xx}/\tau_f$.

A different approach to define $\theta$ would be based on this vertically integrated stress boundary condition in 1HD with $\theta^{1HD} =$

$\tau_{xx}/\tau_f$. In 1HD the normal at the grounding line is equal to the flow direction. In 2HD, this is not necessarily true. Thus, to generalize the longitudinal direction in the 1HD buttressing ratio, a choice needs to be made. The longitudinal direction can either be generalized as the normal at the grounding line ($\theta_2$) or as the flow direction ($\theta_3$).

**Appendix B: Consistent results using different model parameters**

We test the robustness of our findings with respect to the mesh, the sliding law stress exponent $m$, the definition of the

buttressing parameter $\theta$ and the regularization parameter $\gamma_s$. In a second Antarctic setup, based on a different, continent-wide mesh with quadratic base functions (instead of linear elements, see Fig. B1), we find a similar pattern of $\theta_1 \leq 0$ which



yields similar flux differences as exemplified in Fig. B2 for the Filchner-Ronne Ice Shelf. In this case, inversion was done for element-based basal slipperiness and ice softness (instead of inverting on a nodal basis) using a Bayesian methodology (instead of Tikhonov regularization) and the MEASURES velocity data set (Rignot et al. (2011) instead of Landsat 8 (Gardner et al., 2017)). This setup is further described in Reese et al. (2017).

5    For the Antarctic-wide setup described in Sect. 2.2, we test for the choice of the stress exponent $m$ in the sliding law. Different choices $m = 1, 3, 7$ yield good agreement in modeled fluxes but large disagreement in-between analytical fluxes, see Fig. B3. Comparing shelf-wide integrated fluxes for major Antarctic ice shelves shows that also the definitions $\theta_2$ and $\theta_3$ of the buttressing parameter yield large deviations from the modeled fluxes, see Fig. B4. Similarly, we find that the choice of the regularization parameter $\gamma_s$ does not influence the results significantly, see Fig. B5. Our findings are hence independent of the

10  details of numerical modeling choices.

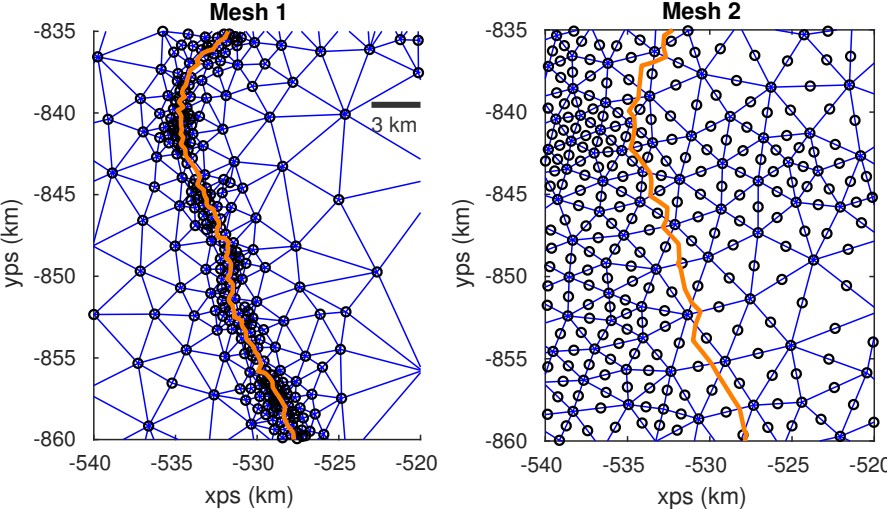

**Figure B1.** Exemplary zoom into the Bindschadler grounding line for two different meshes. Left panel: elements and nodes of the mesh presented in the main text. The mesh was refined especially around the grounding line and linear 3-node elements were employed. Right panel: alternative mesh with 6-node elements with quadratic base functions. The grounding line position is indicated in both meshes in orange.



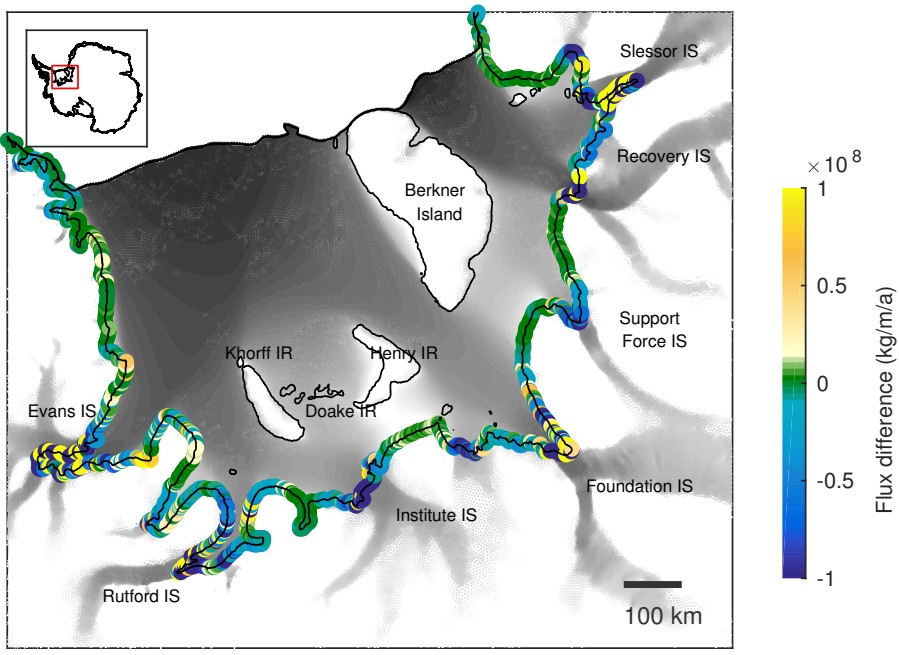

**Figure B2.** Difference between formula-derived and modeled fluxes along the grounding lines of Filchner-Ronne Ice Shelf. In contrast to Fig. 5 a different mesh was employed (exemplified in the right panel in Fig. B1), the data assimilation was conducted using Bayesian inversion and based on the MEASURES velocity data set (Rignot et al., 2011). The analysis was done using quadratic elements. This Antarctic-wide setup is described in more detail in Reese et al. (2017).

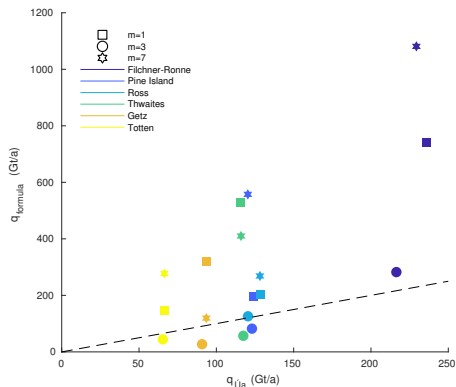

**Figure B3.** Comparison of fluxes calculated with Úa (x-axis) and with the analytical flux formula (y-axis), integrated along the grounding lines of exemplary ice shelves. Symbols indicate the different sliding law exponents $m = 1, 3, 7$ employed. All other parameters agree with the reference run (indicated by a circle). The dotted line shows where fluxes calculated with Úa and predicted by the formula would agree.



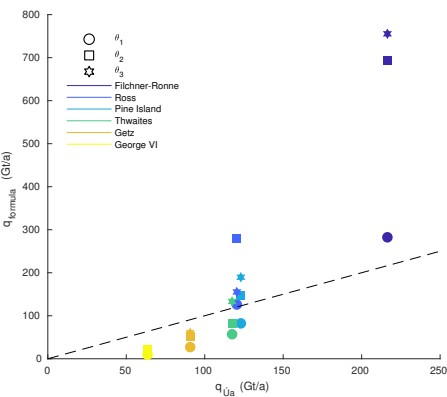

**Figure B4.** Comparison of fluxes calculated with Úa (x-axis) and with the extended flux formula (y-axis), integrated along the grounding lines of exemplary ice shelves. Symbols indicate the different definitions of $\theta$ as described in Sect. 2. All other parameters agree with the reference run (indicated by a circle). The dotted line shows where fluxes calculated with Úa and predicted by the formula would agree.

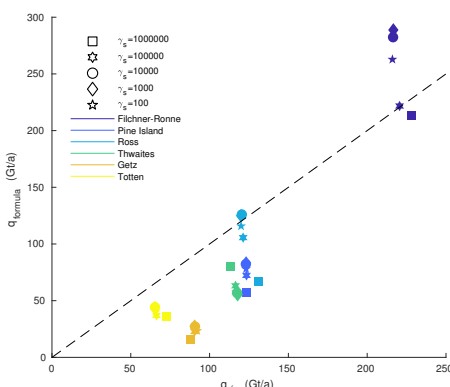

**Figure B5.** Comparison of fluxes calculated with Úa (x-axis) and with the extended flux formula (y-axis), integrated along the grounding lines of exemplary ice shelves. Symbols indicate the different regularization parameters $\gamma_s$ used. All other parameters agree with the reference run (indicated by a circle). The dotted line shows where fluxes calculated with Úa and predicted by the formula would agree.



*Competing interests.* Author G.H. Gudmundsson is a member of the editorial board of the journal.

*Acknowledgements.* The research leading to these results has received funding from the from COMNAP Antarctic Research Fellowship 2016, the German Academic National Foundation, Evangelisches Studienwerk Villigst.



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
