# Peer review of "Grounding-line flux formula applied as a flux condition in numerical simulations fails for buttressed Antarctic ice streams"

_The Cryosphere, 2017_

## Referee Comment (RC1) · Anonymous Referee #1 · 23 Feb 2018

Summary and comments on the manuscript entitled

**Grounding-line flux formula applied as a flux condition in numerical simulations fails for buttressed Antarctic ice streams**

presented on 30.12.2016

by

R. Reese et al.

**Summary**

As reported in several studies, resolution is key for an appropriate representation of grounding line migration in ice-flow models. Yet for continental-scale forward simulation, a trade-off has to be made between resolution and computational costs. On longer time-scales, the mesh has furthermore to be adapted and updated according to the migration of the grounding line. Avoiding these mesh issues, boundary layer theory allows to infer ice-flux values across the grounding line (Schoof, 2007). This flux relation (Eq. 1) has since been implemented as an internal boundary condition in several ice-sheet models using coarse resolution (¿5km). Intercomparison studies confirmed the utility of this parameterisation. Yet the effect of ice-shelf buttressing, which can be included in the parameterisation, did not receive much attention. The authors of this study try to shed light on the applicability of the buttressing correction term (buttressing factor) in the flux parameterisation at the grounding line. For this purpose, a state-of-the-art ice-flow model is applied over Antarctica. Ice velocity observations are matched by a bi-variate inversion for the basal slipperiness and the rate factor. For this setup, it is reported that the computed buttressing factor shows high spatial variability along the grounding line of the two largest ice-shelves in Antarctica. Moreover, negative values are widespread, which leads to unphysical flux values in the parameterisation. The authors therefore strongly question the applicability of this correction term.
The study is soundly structured, to the point and well written. Yet I have major questions and concerns on important details of the methodology and on theit interpretation. As the simultaneous inversion of two parameters is not well posed, their quantification is under-determined. Thus, essential parameters for the flux parameterisation are not well constrained. Further issues concerning the buttressing factor $\theta$ come from uncertainties in the exact location of the grounding line, which might have severe consequences for the interpretation. Therefore, I recommend that this manuscript undergoes a major revision. I advise the editor to only consider publication of this article, if the authors were able to adequately address the comments below.

**Main comments**

**Buttressing factor**

- As you seek to infer flux values at the grounding line, special attention has to be given to its positioning in the model. Errors in the positioning can have large impacts on the flux formula. I therefore highly appreciate the effort to correct the ice density by firn information from a regional climate model. Yet in regions of high inflow from outlet glaciers, the used flotation criterion is possible not very accurate. Certainly in light of the fact that the Bedmap2 thickness values near the grounding line are subject to large uncertainties. For any mismatch of the grounding line position or in ice thickness, the inversion has to compensate by accordingly adjusting the basal slipperiness (C) and the rate factor (A) such that observed velocities are reproduced. I cannot foresee what effects this might have on the stress regime and ultimately on the buttressing parameter ($\theta$).

- Considering uncertainties in the grounding line position, you could rely on Antarctic-wide grounding line observations (e.g. Rignot et al., 2011), rather than using a flotation criterion. Bedmap2 also comprises a flotation mask. A sensitivity assessment to the grounding line position might indeed be valuable.

- You nicely explain how you retrieve the grounding line position. For the calculation of the buttressing factor $\theta$ and the flux parameterisation (Eq. 1), you interpolate the relevant stress and thickness values within the corresponding mesh element. Any grounding-line mesh element will by definition include nodal information from the grounded side. Therefore, your buttressing calculation is biased by a stress regime that is influenced by basal friction. From how I understand the buttressing factor, it should represent the stress regime on the ice shelf-side only because it is this side which exerts buttressing upstream. Unfortunately, this is highly uncharted research territory and there are certainly no best practices. From my experience, some spatial smoothing of the shelf-side deviatoric stress regime is beneficial.

- Using your three buttressing definitions, you compute the 'analytical' parameter $\theta$ all along the grounding line. Yet these values remain uncontested or unvalidated. You concentrate on the fact that negative values are widespread and that the flux parameterisation would yield 'unphysical' results.Buttressing can however be estimated in another way for comparison. The idea is that you should remove the ice shelf entirely and recompute the associated unbuttressed velocity field. Then, you can directly compute the ratio between ice fluxes in the buttressed and unbuttressed case. These values would be very informative to quantify the buttressing effect along the grounding line. Moreover, the variability could be compared to the $\theta$ values.

With some luck the two values show a significant correlation, which you might want to exploit for an improved/adjusted quantification of buttressing in the flux formula.

**Flux formula**

- Similar to above, any uncertainties in the location and ice thickness of the grounding line will affect the inferred parameters (C,A). Moreover, it was shown by (Arthern and Gudmundsson, 2010) that a similar bi-variate inversion for C and A is highly underdetermined. This means that multiple combinations of C and A (on grounded ice) are possible. As both parameters enter the flux parameterisation, with different weighting, these two issues cannot be ignored. I even fear that the usage of the flux parameterisation in such a situation is almost a vain exercise. An option to prove me wrong would be the following: (1) Infer A and C using velocity observations. (2) Remove all ice shelves, either by reducing the computational domain or by setting the ice-shelf thickness to a very small value. Then compute the corresponding velocity field for this unbuttressed case (3) Compare modelled ice flux with the 'analytical' flux parameterisation. I fear that no consistency will be found in the unbuttressed flux values between the 'modelled' and 'analytical' values. This is certainly a very useful and informative exercise.

**Unbuttressed situation**

- I understand why you strongly focus on highly buttressed ice shelves. In this way, your evaluation of the buttressing parameter $\theta$ is deliberately biased to the buttressed cases. I would therefore suggest to add an unconfined ice-shelf setup. An example could be the Thwaites Glacier area, though there might be some complication from a pinning point not present in the Bedmap2 geometry. The pinning point might however not matter to much, as the western portion of the floating tongue is certainly not much buttressed. In such a clearly unconfined setup, I would expect $\theta$ values consistently close to 1.0 or even above.

**Specific comments**

P7L21-23   I suppose that you compute the 'modelled' ice flux over the grounding line by using the velocity component perpendicular to the grounding line. Is that right?

P8L12-13   I do not see the velocity decrease along the central flowline of Institute Ice Stream in Fig. 1. A velocity profile, as an inset to Fig. 1 or 3, would help.

Eq. 14   I wonder how you compute the velocity mismatch in the cost function. Do you do this on the model mesh nodes or directly at the location of each velocity observation. This matters, because you have refined your grid near

the grounding line. So the nodal difference computation would introduce a strong bias in the cost towards the grounding line area. This might even be desirable in your case.

Fig.1    The vectors in this plot are hard to discern. I would prefer a 2D magnitude plot of the velocity fields with superposed streamlines. As mentioned above, an along-flow profile of velocity magnitudes would help to see the velocity decrease upstream of the grounding line.

Fig.3b, 5,B2    Use a different colour map for the flux difference because the colours are very similar to the buttressing parameter $\theta$.

Fig. B3-B5    Legend entries are too tiny. Please take a larger font size.

Fig.S.3.    A comparison of the inferred viscosity field on Ronne Ice Shelf by Larour et al. (2005), I miss well imprinted weak zones along the lateral ice-shelf margins. Do you have any explanation for that?

**References**

Arthern, R. and Gudmundsson, G.: Initialization of ice-sheet forecasts viewed as an inverse Robin problem, Journal of Glaciology, 56, 527–533, doi:{10.3189/002214310792447699}, 2010.

Larour, E., Rignot, E., Joughin, I., and Aubry, D.: Rheology of the Ronne Ice Shelf, Antarctica, inferred from satellite radar interferometry data using an inverse control method, Geophysical Research Letters, 32, doi:{10.1029/2004GL021693}, 2005.

Pattyn, F., Perichon, L., Durand, G., Favier, L., Gagliardini, O., Hindmarsh, R., Zwinger, T., Albrecht, T., Cornford, S., Docquier, D., Fürst, J., Goldberg, D., Gudmundsson, G., Humbert, A., Hütten, M., Huybrechts, P., Jouvet, G., Kleiner, T., Larour, E., Martin, D., Morlighem, M., Payne, A., Pollard, D., Rückamp, M., Rybak, O., Seroussi, H., Thoma, M., and Wilkens, N.: Grounding-line migration in plan-view marine ice-sheet models: results of the *ice2sea* MISMIP3d intercomparison, Journal of Glaciology, 59, 410422, doi:10.3189/2013JoG12J129, 2013.

Rignot, E., Mouginot, J., and Scheuchl, B.: Antarctic grounding line mapping from differential satellite radar interferometry, Geophysical Research Letters, 38, doi:10.1029/2011GL047109, URL http://dx.doi.org/10.1029/2011GL047109, 2011.

Schoof, C.: Ice sheet grounding line dynamics: Steady states, stability, and hysteresis, Journal of Geophysical Research, 112, 19 pp., doi:10.1029/2006JF000664, 2007.

---

## Referee Comment (RC2) · C. Schoof (Referee) · 27 Mar 2018

This paper presents a systematic comparison of remotely-sensed ice fluxes through Antarctic grounding lines with the fluxes predicted by a suite of "flux formulae" based on a particular boundary layer model for ice flow at the grounding line. The basal friction parameter — a key variable in the flux formula — estimated using an inversion of the same velocity data set. In general, the procedure adopted shows terrible agreement, demonstrating that said flux formulae do not work at all well when applied to present-day Antarctica. I believe the result is robust, and I am generally happy for the paper to be published more or less as is.

[Figure]

There are a few items that one could go after a bit more. My view of the review process has become pretty cynical (is my role really to hold up decent work just because it could be improved? Where does that process end?). In short, I am under no illusion that the points I raise will, or even should, feature in a revised paper, and so I have no desire to force the authors to address them. Rather, my argument is that writing off the methodology behind the offending flux fomrulae may be somewhat premature, and this could be investigated further. In that vein, here goes:

1. The discussion regarding the reason for the discrepancy between observed fluxes and fluxes predicted by the flux formulae is a bit weak. The discrepancy is primarily put down to large buttressing factors. I actually think that is inaccurate, in the following sense: if I really had a locally uniaxial flow, and buttressing due to a larger-scale reduction in extensional stress at the grounding line, I'd expect an amended flux formula (with buttressing factor theta_1 as defined in the paper here) to work pretty well, and I wouldn't expect theta_1 = 1/4 to cause major issues. In fact, it had been my impression that Gudmundsson et al (2012) had found reasonably good agreement, at least where the grounding line cuts across the channel in the geometry used in that paper, rather than at its sides (where the flow is presumably heavily affected by shearing parallel to the grounding line).

2. The flux formula being discussed was dervied under a number of conditions (for this you really have to look in detail at Schoof 2007 in JFM, the JGR version won't help), which can really be boiled down as follows: there is a boundary layer near the grounding line over which extensional stress decays to values compatible with a shallow ice approximation further inland. In order for the flux formula being tested to work, the following conditions must be met a) the boundary layer must be in a pseudo-steady state. This is justified by a separation of time scales: the boundary layer should equilibrate much faster than the ice sheet as a whole, so in the absence of rapid changes in forcing (or in bed condition due to internal feedbacks! - see Robel et al 2016 in The Cryosphere) b) the flow at the boundary layer scale must not depend on the transevrse
position (that is, if I move parallel to the grounding line by a distance comparable with the boundary layer length scale, the flow field should still look the same) c) the flow must be unidirectional and perpendicular to the grounding line. The condition that depth-averaged extensional stress ("R") at the grounding line is equal to 1/2 rho(1-rho/rho_w) g H is really not essential at all, which is where the simple correction factor theta in Schoof (2007) came from. It is true that making theta very small should change things - in fact, the extensional stress can become comparable to those that are experienced in a shallow ice flow and the need for a boundary layer almost goes away, as discussed in the appendix to Schoof (2007, JFM), Kowal et al (2013, JFM) as well as the supplementary material to Schoof et al (2017, The Cryosphere). A superficial reading of Reese et al would suggest that the value of theta is the crux of the problem, and I think that obscueres a few things (in the sense that some of the theta values observed, notably the negative ones, are likely to be the symptom rather than the cause; bear with me).

3. Out of the conditions listed above, c) is actually the easiest to surmount (I am currently working on an extension to the unidirectional flow model that would take shearing parallel to the grounding line into account, this changes the relationship between flux, thickness and extensional stress to take account of grounding-line-parallel shear stress, but nothing too exciting happening here). My impression is that the conditions that are most likely to be violated by the real data considered in this paper are a) and b). I'll touch on both below, in a way that may be at least partially testable.

4. Pseudo-steady state (point a above). This is likely to be violated relatively frequently, at least *in the data*. By that, I mean that bedmap ice thicknesses and ice velocities are taken at face value here, and the inversion is done purely as a snapshot inversion to find C, presumably at the cost of generating quite large ice flux divergences, if one were to compute them. If so, then it is likely that a "progonostic" forward computation of the model with time stepping would lead to significant transients that result from an incompatibility between velocity field and bed geometry, and these transients may not be

"real". See the work by Morlighem et al on inverting for bed topography to suppress the effect, and also the work by Goldberg and Heimbach on the use of data assimiliation techniques to avoid the pitfalls of using snapshot inversions for bed properties.

From the MISMIP model intercomparison in one horizontal dimension (Pattyn et al 2012, The Cryosphere), we know that violating the pseduo-steady state assumption (in that case, due to step changes in the ice viscosity parameter) can lead to singificant but sort-lived departures from the "flux formula", with the ice flux at the grounding line potentially settling back onto the flux formula over a time scale that is equal to the advective time scale for the boundary layer (boundary layer length / scale for velocity in the boundary layer). I'm not saying that this is likely to happen here, but it's worth illustrating - if you run your model forward over the time scale I identify, do you end up somewhere closer to the flux formula?

A corollary of this is actually the negative theta values computed. If I supposed that the flow were unidirectional and laterally homogeneous as per points c) and b) above, then the only way I could have a negative theta value would be if $du/dx < 0$ (u and x being measured along the flowline, naturally. If that is the case, then the assumption of the boundary layer model of a fixed flux through the boundary layer, corresponding to a pseudo-steady state, must fail, since such a flux would require $d(hu)/dx = h\, du/dx + u\, hd/dx = 0$ and we can assume that u, h > 0 and, as we ought to be thinning towards the grounding line, $dh/dx < 0$. If also $du/dx < 0$, both terms are negative and there is no way that we can have $h\, du/dx + u\, hd/dx = 0$.

That leads to two possibliities: either the pseudo-steady state condition is violated (so a) does not apply, which could occur due to short-time-scale changes in forcing) or the assumption of laterally homogeneous flow (point b) above) must be false. I expect we are looking at the latter rather than the former: the flow of the ice streams in question slows as the grounding line is approached. Mass is not lost here, but rather, I expect that the flow simply spreads laterally: as the centreline slows, a wider region flows at ice stream speeds. I confess I haven't bothered to check this in the data set, but it
would certainly be worth doing that

5. Lateral homogeneity. This is probably the big one. The original boundary layer theory did not deal with this at all, and I fully expect that ice streams violate this assumption pretty much every time. There is plenty of evidence for the force balance of ice streams to be singificantly affected by lateral shear stresse, and I expect this holds true near the grounding line. This is intrinsically a loss of lateral homogeneity: we have gradients in the velocity component normal to the grounding line with respect to the coordinate that measures distance parallel to the grounding line, and an extra term appears in the force balance that does not feature in the original Schoof (2007, JFM) boundary layer theory. In that sense, I would not expect the resutls of the latter paper to hold, though I know how one would update the boundary layer theory presented there to account for it. In fact, it may be worth pointing out that various papers have already tried to do that, though only in a form where lateral drag is parameterized. The most complete version of this can be found in Schoof et al (2017, The Cryosphere), with parts of the problem also addresed by Pegler (2016 and earlier papers).

Is this testable? The answer is yes. Both Schoof (2007) papers give estimates for the size of the boundary layer as a function of the various model parameters, such as C, m etc. All you would have to do is check whether any mdoel parameters (such as extensional stess at the grounding line, or C, or ice thickness at the grounding line) vary singificantly within that boundary layer length scale. It is not acutally enough to do that *along* the grounding line, you should really also check that C in particular does not vary by an O(1) fractional amount when going a single boundary layer length scale inland. If there is singificant variability, you immediately have good reason to say that the flux formula won't hold.

6. I'm not particularly bothered by wanting to "defend" simulation models that use a flux formula. I can ceratinly see their appeal in simulating long time scales, and for testing qualititative ice sheet behaviour without getting lost in computational detail. But I haven't built a career on such a model. That said, I would be quite interested in a com-

parison of long-time-scale prognostic modelling using a flux formula model - admitting that it is going to give locally terribly wrong fluxes at a given point in time - and a fully resolved model like Ua. If I care about grounded ice volume changes over long time scales, how different are the predicitons? And are they *qualitiatively* different, in the sense that irreversible retreat occurs not just at different values of forcing parameters, but depends in a fundamentally different way on the forcing that is imposed?

Minor comments:

"Within the context of the shallow ice-stream computational models — a commonly-used flow approximation for describing the flow of ice streams and ice shelves (e.g., Morland, 1987; MacAyeal, 1989) — it has, for example, been suggested that for many applications a horizontal resolution of around one ice thickness or less is suitable (Gladstone et al., 2012; Pattyn et al., 2012; Cornford et al., 2016)" This is a somewhat bizarre thing to say; the whole point about a shallow ice theory is that it does not know about how long a horizontal distance equal to one ice thickness is: the limit of a small aspect ratio is already implicit in constructing a shallow ice theory. The fact that a grid or node spacing comparable to something like 1-3 km (or whatever) is regarded as adequate should not be equated with a mesh element size of one ice thickness. More relevant is what fraction of the linear domain size the typical distance between grid points or nodes should be — probably 1 in 1000 is really implied here.

equation 14 "I(f) = ..." what does the argument "f" signify, given that f does not appear on the right-hand side of equation (14)? Should this be I(v)?

page 9 "can \emph{not} be used": this should be "\emph{cannpt} be used"

figure 4: The Ua flux looks terrible - as in, very non-smooth. I'd expect H1-type convergence of ice velocities under grid refinement - is the jaggedness mostly a result of a misalignment between grounding line and mesh, or of forcing the grounding line to lie along a mesh (and therefore having sharp corners at every node?) Probably worth explaining.

page 13 bottom "However, in the presence of ice-shelf buttressing no such simple conclusions can be drawn (e.g Goldberg et al., 2009; Gudmundsson et al. ,2012; Gudmundsson, 2013; Pegler, 2016)." Without wishing to advertise my own work too much, Schoof et al 2017 in the Cryosphere also gives a qualitatively different example (with calving, which I believe differs from the other references given here) where the usual stability argument is reversed.

multiple instances, for instance figure caption B1. "Exemplary" is not usually used synonymously with "an example of". A brief internet search gives me the following meanings 1. serving as a desirable model; representing the best of its kind. "an award for exemplary community service" synonyms: perfect, ideal, model, faultless, flawless, impeccable, irreproachable; More excellent, outstanding, admirable, commendable, laudable, above/beyond reproach; textbook, consummate, archetypal "her exemplary behavior" antonyms: deplorable 2. (of a punishment) serving as a warning or deterrent. "exemplary sentencing may discourage the ultraviolent minority" synonyms: deterrent, cautionary, warning, admonitory; raremonitory "exemplary jail sentences" You might want to consider whether this is what you mean

Christian Schoof

---

## Author Comment (AC1) · 23 May 2018

Please find our response to all reviewer comments in the attached pdf. All changes in the manuscript (generated by latexdiff) can be found at the end of the file.

Please also note the supplement to this comment:
https://www.the-cryosphere-discuss.net/tc-2017-289/tc-2017-289-AC1-supplement.pdf

---

## Author Response (AR1)

**Response to Reviewer Comments**
**Date: 20 April 2018**
**By R. Reese, R. Winkelmann, G. H. Gudmundsson**

Journal: TC

Title: Grounding-line flux formula applied as a flux condition in numerical simulations fails for buttressed Antarctic ice streams

Author(s): R. Reese, R. Winkelmann, G. H. Gudmundsson

MS No.: tc-2017-289

MS Type: Research article

First of all, we would like to thank the editor Olivier Gagliardini, the anonymous reviewer and Christian Schoof for their helpful and excellent comments and their efforts to create the detailed reviews!

In our response and in the revision of the manuscript we addressed the main issues raised by the first reviewer:

1. We detailed the description of our inverse methodology (see, e.g., comments on 'flux formula' and first comment on the 'buttressing factor') and created figures to underpin that our findings are independent of the details of the inversion as requested (Figs. 2, 1).

2. We discuss that the exact grounding line location does not affect our results (see, e.g., second comment on 'buttressing factor', lines 4f on page 19 of the marked-up manuscript, and Figs. 2, 1 here).

We are further very happy about the evaluation of our manuscript by Christian Schoof and would like to thank him for his discussion about the underlying reasons for the formula to fail at Antarctic grounding lines. We would be very happy to collaborate with him along the lines of his suggestions to approach this task in an further study.

We provide detailed answers to all comments below. The reviewer's comments are given in black and the authors responses in blue. The changes made to the main document can be found at the end (created with latexdiff). Page and line numbers given below relate to this document.

**Anonymous referee 1**

The comment was uploaded in the form of a supplement:
https://www.the-cryosphere-discuss.net/tc-2017-289/tc-2017-289-RC1-supplement.pdf

**Summary**
As reported in several studies, resolution is key for an appropriate representation of grounding line migration in ice-flow models. Yet for continental-scale forward simulation, a trade-off has to be made between resolution and computational costs. On longer time-scales, the mesh has furthermore to be adapted and updated according to the migration of the grounding line. Avoiding these mesh issues, boundary layer theory allows to infer ice-flux values across the grounding line (Schoof, 2007). This flux relation (Eq. 1) has since been implemented as an internal boundary condition in several ice-sheet models using coarse resolution ($\sim$ 5km). Inter-comparison studies confirmed the utility of this parameterisation. Yet the effect of ice-shelf buttressing, which can be included in the parameterisation, did not receive much attention. The authors of this study try to shed light on the applicability of the buttressing correction term (buttressing factor) in the flux parameterisation at the grounding line. For this purpose, a state-of-the-art ice-flow model is applied over Antarctica. Ice velocity observations are matched by a bi-variate inversion for the basal slipperiness and the rate factor. For this setup, it is reported that the computed buttressing factor shows high spatial variability along the grounding line of the two largest ice-shelves in Antarctica. Moreover, negative values are widespread, which leads to unphysical flux values in the parameterisation. The authors therefore strongly question the applicability of this correction term. The study is soundly structured, to the point and well written. Yet I have major questions and concerns on important details of the methodology and on theit interpretation. As the simultaneous inversion of two parameters is not well posed, their quantification is underdetermined. Thus, essential parameters for the flux parameterisation are not well constrained. Further issues concerning the buttressing factor $\theta$ come from uncertainties in the exact location of the grounding line, which might have severe consequences for the interpretation. Therefore, I recommend that this manuscript undergoes a major revision. I advise the editor to only consider publication of this article, if the authors were able to adequately address the comments below.

We would like to thank the anonymous reviewer for his/her effort to review our manuscript and appreciate his/her comments for improving our study. His/Her main points are raised on the inversion method to initialize our model and the positioning of the grounding line in the study. We give in-depth responses to both issues that the reviewer detailed in the questions below (see comments on 'buttressing factor' and 'flux formula').
However, we would like to point out that $\theta$ becomes negative independently of the initialization

method of our study and of the exact grounding line position in the setup: In Institute Ice Stream, this fact is already visible in the velocity data set. Ice velocities decrease approaching the grounding line (see Fig. 1 below, also added to Fig. 1 of the main manuscript). Negative, longitudinal strain rates are the underlying reason for the buttressing value to become negative. Similar strain rates are found within a region that extends 20km upstream and 100km downstream of the grounding line. Negative strain rates in longitudinal direction relate to compressive stresses and $\theta \leq 0$, as described on page 10 in lines 15ff of the marked-up manuscript. Hence, $\theta \leq 0$ is independent of both main issues raised by the reviewer: (1) the inversion methodology to determine the basal slipperiness and ice rate factor and (2) the exact position of the grounding line in the model experiments. Unrelated to both, the formula will hence fail in this case.

[Figure]

**Figure 1:** Speed along the centerline of Institute Ice Stream. Observed and modelled velocities decrease towards the grounding line.

**Main comments**
**Buttressing factor**

- As you seek to infer flux values at the grounding line, special attention has to be given to its positioning in the model. Errors in the positioning can have large impacts on the flux formula. I therefore highly appreciate the effort to correct the ice density by firn information from a regional climate model. Yet in regions of high inflow from outlet glaciers, the

used flotation criterion is possible not very accurate. Certainly in light of the fact that the Bedmap2 thickness values near the grounding line are subject to large uncertainties. For any mismatch of the grounding line position or in ice thickness, the inversion has to compensate by accordingly adjusting the basal slipperiness ($C$) and the rate factor ($A$) such that observed velocities are reproduced. I cannot foresee what effects this might have on the stress regime and ultimately on the buttressing parameter ($\theta$).

We agree with the reviewer that the position of the grounding line needs particular care, especially in respect with the uncertainties in the Bedmap2 data set. That is the reason why we adjusted the Bedmap2 bedrock data within a range of 50m so that the grounding line position aligns with the observed position from (Bindschadler et al., 2011). This is described on page 7 in lines 7f in the marked-up manuscript. Thus, the modelled grounding line position, which is determined from the physically-based flotation criterion, matches closely observed grounding lines. However, the results of our study are independent of the exact position of the grounding line. As Fig. 1 below as well as Fig. 3 in the main manuscript show, velocities decrease as the ice approaches the grounding line, which gives rise to the negative $\theta$ values along the grounding line of Institute Ice Stream. This pattern is not restricted to the grounding line exclusively, but extends also upstream and downstream of the grounding line and will hence not depend on the exact location of the grounding line in that region. This is already discussed on page 16, lines 22ff as well as page 10, line 32ff of the manuscript.

- Considering uncertainties in the grounding line position, you could rely on Antarctic-wide grounding line observations (e.g. Rignot et al. (2011a)), rather than using a flotation criterion. Bedmap2 also comprises a flotation mask. A sensitivity assessment to the grounding line position might indeed be valuable.

As stated above, we make use of the observed grounding line position from Bindschadler et al. (2011) in order to adjust the bed topography around the grounding line. The adjustment of the bed topography within the range of Bedmap2 uncertainties was done only for the Antarctic-wide setup presented in the main manuscript, for the alternative mesh, described in the Appendix B and shown in Figs. B1 and B2, this procedure was not applied. While its grounding line position hence deviates from the observed positions, the results are qualitatively and quantitatively very similar and we conclude that our findings are robust with respect to the grounding line position. We added this on page 19, line 4f of the marked-up manuscript.

- You nicely explain how you retrieve the grounding line position. For the calculation of the buttressing factor $\theta$ and the flux parameterisation (Eq. 1), you interpolate the relevant

stress and thickness values within the corresponding mesh element. Any grounding-line mesh element will by definition include nodal information from the grounded side. Therefore, your buttressing calculation is biased by a stress regime that is influenced by basal friction. From how I understand the buttressing factor, it should represent the stress regime on the ice shelf-side only because it is this side which exerts buttressing upstream. Unfortunately, this is highly uncharted research territory and there are certainly no best practices. From my experience, some spatial smoothing of the shelf-side deviatoric stress regime is beneficial.

We do not fully understand this comment and hope that we can address it appropriately: we calculate the buttressing number from the stress fields at the grounding line position diagnosed from the floatation criterion. Since stresses within the ice are continuous across the grounding line using an interpolation method is valid here. A closer look into the stress fields (see Fig. 2) reveals that it is rather uniform in the vicinity of the grounding line and hence the stresses are independent of the exact position of the grounding line and of the interpolation method.

- Using your three buttressing definitions, you compute the 'analytical' parameter $\theta$ all along the grounding line. Yet these values remain uncontested or unvalidated. You concentrate on the fact that negative values are widespread and that the flux parameterisation would yield 'unphysical' results. Buttressing can however be estimated in another way for comparison. The idea is that you should remove the ice shelf entirely and recompute the associated unbuttressed velocity field. Then, you can directly compute the ratio between ice fluxes in the buttressed and unbuttressed case. These values would be very informative to quantify the buttressing effect along the grounding line. Moreover, the variability could be compared to the $\theta$ values. With some luck the two values show a significant correlation, which you might want to exploit for an improved adjusted quantification of buttressing in the flux formula.

While we agree that the effect of buttressing could also be quantified by relating modelled fluxes in the presence and absence of an ice shelf, it is unclear to us how such an experiment relates to the topic of our manuscript, which is to test the accuracy of the flux formula under realistic conditions. In particular, it is unclear how this would produce $\theta$ values of relevance for calculating analytical ice fluxes. As there appears to be some misunderstanding on the behalf of the reviewer as to how $\theta$ is defined in the original work, it is done by replacing condition (9) with $2\bar{A}^{-1/n}h|\frac{\partial u}{\partial x}|^{1/n-1}\frac{\partial u}{\partial x} = \theta \cdot \left(\frac{1}{2}\rho_i\left(1 - \frac{\rho_i}{\rho_w}\right)gh^2\right)$ at $x = x_{gl}$ in order to account for modification in stress at the grounding line from the 1HD unbuttressed case (see description in Section 4.2 of (Schoof, 2007)).
The reviewer appears to suggest that if $\theta$ was somehow defined or calculated in a different way, the analytical fluxes would show a better agreement with calculated ones (but provides no evidence to support this). As we explain in the manuscript several slightly different definitions of $\theta$ have been used in the literature. While our first definition of $\theta$ (i.e. $\theta_1$) is in our opinion the correct definition, we found after having conducted a thorough review of the literature that two further definitions have also been used. We therefore decided to conduct our analysis using all three definitions of $\theta$. In all cases our conclusions are the same: The flux formula produces either grossly incorrect or quite simply physically unrealistic fluxes for all major Antarctic ice streams.

[Figure]

**Figure 2:** Close-up of the grounding line of Institute Ice Stream from Fig. 3a in the main manuscript (location indicated in the inset).

**Flux formula**

- Similar to above, any uncertainties in the location and ice thickness of the grounding line will affect the inferred parameters $(C, A)$. Moreover, it was shown by (Arthern and Gudmundsson, 2010) that a similar bi-variate inversion for $C$ and $A$ is highly underdetermined. This means that multiple combinations of C and A (on grounded ice) are possible. As both parameters enter the flux parameterisation, with different weighting, these two issues cannot be ignored. I even fear that the usage of the flux parameterisation in such a situation is almost a vain exercise. An option to prove me wrong would be the following: (1) Infer $A$ and $C$ using velocity observations. (2) Remove all ice shelves, either by reducing the computational domain or by setting the ice-shelf thickness to a very small value. Then compute the corresponding velocity field for this unbuttressed case (3) Compare modelled

ice flux with the 'analytical' flux parameterisation. I fear that no consistency will be found in the unbuttressed flux values between the 'modelled' and 'analytical' values. This is certainly a very useful and informative exercise.

We have some difficulties understanding the purpose of the proposed exercise. In the unbuttressed case the analytical and the numerical fluxes will (of course) agree. This has been tested many times in the past both with our ice-flow model Úa, and with other similar models. There seems to be little use in repeating this exercise here. However we would like to point out that simply removing ice shelves will not provide a flowline-type unbuttressed situation as the reviewer seems to suggest, as even in that situation the flow can be convergent/divergent at the ice margin, in which case the buttressing parameter $\theta$ will not be equal to unity. We would also like to point out that the example given in Arthern and Gudmundsson, 2010, referred to a datum flow where there are no spatial variations in ice or bed properties. The more general situation (i.e. where A and C vary spatially) has yet to be studied, but we expect the results of such a study to be somewhat similar to the conclusions by Gudmundsson and Raymond, 2008, who studied a similar type of an inverse problem involving spatial variations in both basal slipperiness and basal topography. There it was found that the problem was, giving the right surface data, not necessarily under-determined. We did test for a number of different degrees of regularization as well as different values of $m$, which yield different $A$ and $C$ fields. We find our results to be robust, with details given in Appendix B. Again, we would like to stress that the reason for $\theta \leq 0$ in the region of Institute Ice Stream is already visible in the observed flow fields. This is true for observed fields from (Gardner et al., 2017) as well as Rignot et al. (2011b). The simplest explanation for the differences between the numerically and analytically calculated fluxes is that not all the assumptions made to derive the formula are satisfied. An in-depth discussion on these assumptions is given by Reviewer 2.

**Unbuttressed situation**

- I understand why you strongly focus on highly buttressed ice shelves. In this way, your evaluation of the buttressing parameter $\theta$ is deliberately biased to the buttressed cases. I would therefore suggest to add an unconfined ice-shelf setup. An example could be the Thwaites Glacier area, though there might be some complication from a pinning point not present in the Bedmap2 geometry. The pinning point might however not matter to much, as the western portion of the floating tongue is certainly not much buttressed. In such a clearly unconfined setup, I would expect $\theta$ values consistently close to 1.0 or even above.

In our study, we focus on the current Antarctic Ice Sheet. Since most of its major ice streams and glaciers are highly buttressed, we believe that focusing on buttressed ice streams is of particular importance. Figs. 1 from (Fürst et al., 2016) and (Reese et al.,

2017) show that all Antarctic ice shelves buttress upstream ice flow. In Table 1 of the manuscript we compare the modelled to formula-predicted fluxes and find that also for the (probably) less buttressed grounding line of Thwaites glacier, fluxes disagree by more than 50%. This is true for all major ice shelves. We added a Figure for West Ice Shelf (see Fig. 3, as the reviewer points out in Thwaites a pinning point is missing and we hence do not fully trust the results for Thwaites). The western part of West Ice Shelf is largely unconfined, and $\theta$ values are generally close to 1, but since the grounding line is not perfectly straight, they vary locally and in some areas, $\theta$ values are very low. Fluxes here do not show perfect agreement, which might be pinned down to the fact that one of the assumptions in the derivation of the formula fails as discussed by the second reviewer Christian Schoof. We tested our calculations and the model also for a flowline setup where we find very good agreement between the formula and the modelled fluxes.

[Figure]

**Figure 3:** Same as Figure 3 in the paper, here for West Ice Shelf.

**Specific comments**

P7 L21-23  I suppose that you compute the 'modelled' ice flux over the grounding line by using the velocity component perpendicular to the grounding line. Is that right?
Yes.

P8 L12-13  I do not see the velocity decrease along the central flowline of Institute Ice Stream in Fig. 1. A velocity profile, as an inset to Fig. 1 or 3, would help.
Please find the velocity profile in Fig. 1, also as an inset to Fig. 1 of the main manuscript.

Eq. 14  I wonder how you compute the velocity mismatch in the cost function. Do you do this on the model mesh nodes or directly at the location of each velocity observation. This matters, because you have refined your grid near the grounding line. So the nodal difference computation would introduce a strong bias in the cost towards the grounding line area.

This might even be desirable in your case.

We are not sure that we understand this comment correctly. The misfit function is given as an integral over velocity mismatches. The integral is numerically evaluated at nodes of the mesh. But by integrating over the entire domain, the areas of the respective elements are taken into account and the misfit function is not biased towards the highly-resolved regions of the mesh.

Fig. 1 The vectors in this plot are hard to discern. I would prefer a 2D magnitude plot of the velocity fields with superposed streamlines. As mentioned above, an along-flow profile of velocity magnitudes would help to see the velocity decrease upstream of the grounding line.

Done.

Fig. 3b, 5, B2 Use a different colour map for the flux difference because the colours are very similar to the buttressing parameter $\theta$.

Done.

Fig. B3-5 Legend entries are too tiny. Please take a larger font size.

Done.

Fig. S3 A comparison of the inferred viscosity field on Ronne Ice Shelf by Larour et al. (2005), I miss well imprinted weak zones along the lateral ice-shelf margins. Do you have any explanation for that?

We did not invert for the viscosity but for the rate factore $A$, and there are clearly defined bands of weak ice (i.e. high A values) in the vicinity of most margins.

**Referee 2: Christian Schoof**

This paper presents a systematic comparison of remotely-sensed ice fluxes through Antarctic grounding lines with the fluxes predicted by a suite of "flux formulae" based on a particular boundary layer model for ice flow at the grounding line. The basal friction parameter - a key variable in the flux formula - estimated using an inversion of the same velocity data set. In general, the procedure adopted shows terrible agreement, demonstrating that said flux formulae do not work at all well when applied to present-day Antarctica. I believe the result is robust, and I am generally happy for the paper to be published more or less as is.

We would like to thank you for your efforts and your in-depth, detailed discussion on the underlying reasons for the formula to fail along Antarctic grounding lines. We are happy that you agree with our main conclusion that the applicability of the formula to Antarctic grounding lines is limited. As you mention, $\theta \leq 0$ is more of a symptom and the underlying reason for the formula to fail is that one of the assumptions made to derive the formula is not satisfied for the Antarctic grounding lines. These assumptions are (a) that the boundary layer is in quasi-steady state, (b) that the flow is transversally homogeneous and (c) the flow is unidirectional and perpendicular to the grounding line. We added a remark in the main text to clarify that we do not focus on the underlying reasons here (page 16, lines 10f and page 17, line 13) and we addressed all minor changes (see also specific comments below).

We want to thank you for pointing out various tests to understand which of those conditions fails and we would be very happy to collaborate with you in future along those lines. In this manuscript, we aim to make the point that the amended formula does not work for Antarctic ice streams, independently from the underlying reasons for it to fail, and we are happy that you recommend the paper to be published in its current state.

There are a few items that one could go after a bit more. My view of the review process has become pretty cynical (is my role really to hold up decent work just because it could be improved? Where does that process end?). In short, I am under no illusion that the points I raise will, or even should, feature in a revised paper, and so I have no desire to force the authors to address them. Rather, my argument is that writing off the methodology behind the offending flux formulae may be somewhat premature, and this could be investigated further. In that vein, here goes:

1. The discussion regarding the reason for the discrepancy between observed fluxes and fluxes predicted by the flux formulae is a bit weak. The discrepancy is primarily put down to large buttressing factors. I actually think that is inaccurate, in the following sense: if I really had

a locally uniaxial flow, and buttressing due to a larger-scale reduction in extensional stress at the grounding line, I'd expect an amended flux formula (with buttressing factor $\theta_1$ as defined in the paper here) to work pretty well, and I wouldn't expect $\theta_1 = 1/4$ to cause major issues. In fact, it had been my impression that Gudmundsson et al. (2012) had found reasonably good agreement, at least where the grounding line cuts across the channel in the geometry used in that paper, rather than at its sides (where the flow is presumably heavily affected by shearing parallel to the grounding line).

2. The flux formula being discussed was derived under a number of conditions (for this you really have to look in detail at Schoof 2007 in JFM, the JGR version won't help), which can really be boiled down as follows: there is a boundary layer near the grounding line over which extensional stress decays to values compatible with a shallow ice approximation further inland. In order for the flux formula being tested to work, the following conditions must be met a) the boundary layer must be in a pseudo-steady state. This is justified by a separation of time scales: the boundary layer should equilibrate much faster than the ice sheet as a whole, so in the absence of rapid changes in forcing (or in bed condition due to internal feedbacks! - see Robel et al 2016 in The Cryosphere) b) the flow at the boundary layer scale must not depend on the transevrse position (that is, if I move parallel to the grounding line by a distance comparable with the boundary layer length scale, the flow field should still look the same) c) the flow must be unidirectional and perpendicular to the grounding line. The condition that depth-averaged extensional stress ('$R$') at the grounding line is equal to $1/2rho(1 - rho/rho_w)gH$ is really not essential at all, which is where the simple correction factor theta in Schoof (2007) came from. It is true that making theta very small should change things - in fact, the extensional stress can become comparable to those that are experienced in a shallow ice flow and the need for a boundary layer almost goes away, as discussed in the appendix to Schoof (2007, JFM), Kowal et al (2013, JFM) as well as the supplementary material to Schoof et al (2017, The Cryosphere). A superficial reading of Reese et al would suggest that the value of theta is the crux of the problem, and I think that obscueres a few things (in the sense that some of the theta values observed, notably the negative ones, are likely to be the symptom rather than the cause; bear with me).

3. Out of the conditions listed above, c) is actually the easiest to surmount (I am currently working on an extension to the unidirectional flow model that would take shearing parallel to the grounding line into account, this changes the relationship between flux, thickness and extensional stress to take account of grounding-line-parallel shear stress, but nothing too exciting happening here). My impression is that the conditions that are most likely to be violated by the real data considered in this paper are a) and b). I'll touch on both below, in a way that may be at least partially testable.

4. Pseudo-steady state (point a above). This is likely to be violated relatively frequently, at

least *in the data*. By that, I mean that bedmap ice thicknesses and ice velocities are taken at face value here, and the inversion is done purely as a snapshot inversion to find C, presumably at the cost of generating quite large ice flux divergences, if one were to compute them. If so, then it is likely that a "progonostic" forward computation of the model with time stepping would lead to significant transients that result from an incompatibility between velocity field and bed geometry, and these transients may not be 'real'. See the work by Morlighem et al on inverting for bed topography to suppress the effect, and also the work by Goldberg and Heimbach on the use of data assimiliation techniques to avoid the pitfalls of using snapshot inversions for bed properties.

From the MISMIP model intercomparison in one horizontal dimension (Pattyn et al 2012, The Cryosphere), we know that violating the pseduo-steady state assumption (in that case, due to step changes in the ice viscosity parameter) can lead to singificant but sort-lived departures from the 'flux formula', with the ice flux at the grounding line potentially settling back onto the flux formula over a time scale that is equal to the advective time scale for the boundary layer (boundary layer length / scale for velocity in the boundary layer). I'm not saying that this is likely to happen here, but it's worth illustrating - if you run your model forward over the time scale I identify, do you end up somewhere closer to the flux formula?

A corollary of this is actually the negative theta values computed. If I supposed that the flow were unidirectional and laterally homogeneous as per points c) and b) above, then the only way I could have a negative theta value would be if $du/dx < 0$ ($u$ and $x$ being measured along the flowline, naturally. If that is the case, then the assumption of the boundary layer model of a fixed flux through the boundary layer, corresponding to a pseudo-steady state, must fail, since such a flux would require $d(hu)/dx = hdu/dx + uhd/dx = 0$ and we can assume that $u, h > 0$ and, as we ought to be thinning towards the grounding line, $dh/dx < 0$. If also $du/dx < 0$, both terms are negative and there is no way that we can have $hdu/dx + uhd/dx = 0$.

That leads to two possibliities: either the pseudo-steady state condition is violated (so a) does not apply, which could occur due to short-time-scale changes in forcing) or the assumption of laterally homogeneous flow (point b) above) must be false. I expect we are looking at the latter rather than the former: the flow of the ice streams in question slows as the grounding line is approached. Mass is not lost here, but rather, I expect that the flow simply spreads laterally: as the centreline slows, a wider region flows at ice stream speeds. I confess I haven't bothered to check this in the data set, but it would certainly be worth doing that.

5. Lateral homogeneity. This is probably the big one. The original boundary layer theory did not deal with this at all, and I fully expect that ice streams violate this assumption pretty much every time. There is plenty of evidence for the force balance of ice streams to be singificantly affected by lateral shear stresse, and I expect this holds true near the grounding line. This is intrinsically a loss of lateral homogeneity: we have gradients in the velocity component normal

to the grounding line with respect to the coordinate that measures distance parallel to the grounding line, and an extra term appears in the force balance that does not feature in the original Schoof (2007, JFM) boundary layer theory. In that sense, I would not expect the resutls of the latter paper to hold, though I know how one would update the boundary layer theory presented there to account for it. In fact, it may be worth pointing out that various papers have already tried to do that, though only in a form where lateral drag is parameterized. The most complete version of this can be found in Schoof et al (2017, The Cryosphere), with parts of the problem also addresed by Pegler (2016 and earlier papers).

Is this testable? The answer is yes. Both Schoof (2007) papers give estimates for the size of the boundary layer as a function of the various model parameters, such as C, m etc. All you would have to do is check whether any mdoel parameters (such as extensional stess at the grounding line, or C, or ice thickness at the grounding line) vary singificantly within that boundary layer length scale. It is not acutally enough to do that *along* the grounding line, you should really also check that C in particular does not vary by an O(1) fractional amount when going a single boundary layer length scale inland. If there is singificant variability, you immediately have good reason to say that the flux formula won't hold.

6. I'm not particularly bothered by wanting to "defend" simulation models that use a flux formula. I can ceratinly see their appeal in simulating long time scales, and for testing qualititative ice sheet behaviour without getting lost in computational detail. But I haven't built a career on such a model. That said, I would be quite interested in a comparison of long-time-scale prognostic modelling using a flux formula model - admitting that it is going to give locally terribly wrong fluxes at a given point in time - and a fully resolved model like Ua. If I care about grounded ice volume changes over long time scales, how different are the predicitons? And are they *qualitiatively* different, in the sense that irreversible retreat occurs not just at different values of forcing parameters, but depends in a fundamentally different way on the forcing that is imposed?

Minor comments:

- "Within the context of the shallow ice-stream computational models — a commonly-used flow approximation for describing the flow of ice streams and ice shelves (e.g., MacAyeal, 1989; Morland, 1987) — it has, for example, been suggested that for many applications a horizontal resolution of around one ice thickness or less is suitable (Cornford et al., 2016; Gladstone et al., 2012; Pattyn et al., 2012)." This is a somewhat bizarre thing to say; the whole point about a shallow ice theory is that it does not know about how long a horizontal distance equal to one ice thickness is: the limit of a small aspect ratio is already implicit in constructing a shallow ice theory. The fact that a grid or node spacing comparable to something like 1-3 km (or whatever) is regarded as adequate should not be equated with a mesh element size of one ice thickness. More relevant is what fraction of the linear domain

size the typical distance between grid points or nodes should be - probably 1 in 1000 is really implied here.

Many thanks for pointing this out. We changed the wording to '..it has, for example, been suggested that for many applications a horizontal resolution of around one kilometer or less is suitable...'.

- equation 14 "$I(f) = ...$" what does the argument "$f$" signify, given that f does not appear on the right-hand side of equation (14)? Should this be $I(v)$?

  Thanks for bringing this typo to our attention. The argument should indicate the data misfit $v_{modeled} - v_{observed}$, which we omit now for simplicity.

- page 9 "can *not* be used": this should be "*cannot* be used"

  Done.

- figure 4: The Ua flux looks terrible - as in, very non-smooth. I'd expect H1-type convergence of ice velocities under grid refinement - is the jaggedness mostly a result of a misalignment between grounding line and mesh, or of forcing the grounding line to lie along a mesh (and therefore having sharp corners at every node?) Probably worth explaining.

  This is a discretization issue, depending on the diagnostic location of the grounding line. We diagnose it as a curve of line-segments crossing the (triangular) elements with floating and grounded nodes, with an example given in Fig. B1, Panel 2. This creates a 'wiggly' grounding line. To obtain fluxes, we calculate the grounding line normal for each such segment which causes an element-by-element variation of the normal, and in fluxes. Especially in areas where the flow direction is more or less aligned with the grounding line, this leads to positive-to-negative fluctuations in fluxes. We added this discussion to our manuscript on page 13, lines 1 and 2. If we apply a moving average smoothing to the data (in Fig. 4 displayed for a span of 5 neighbours) Úa fluxes look much better, but we felt better to show the element-wise results.

- page 13 bottom "However, in the presence of ice-shelf buttressing no such simple conclusions can be drawn (e.g Goldberg et al., 2009; Gudmundsson et al. ,2012; Gudmundsson, 2013; Pegler, 2016)." Without wishing to advertise my own work too much, Schoof et al 2017 in the Cryosphere also gives a qualitatively different example (with calving, which I believe differs from the other references given here) where the usual stability argument is reversed.

  We added this study to our manuscript.

- multiple instances, for instance figure caption B1. "Exemplary" is not usually used synonymously with "an example of". A brief internet search gives me the following meanings 1. serving as a desirable model; representing the best of its kind. "an award for

exemplary community service" synonyms: perfect, ideal, model, faultless, flawless, impeccable, irreproachable; More excellent, outstanding, admirable, commendable, laudable, above/beyond reproach; textbook, consummate, archetypal "her exemplary behavior" antonyms: deplorable 2. (of a punishment) serving as a warning or deterrent. "exemplary sentencing may discourage the ultraviolent minority" synonyms: deterrent, cautionary, warning, admonitory; raremonitory "exemplary jail sentences" You might want to consider whether this is what you mean

Thanks for bringing this to our attention. We changed the wording accordingly.

[Figure]

**Figure 4:** Comparison of fluxes calculated with Úa (blue) and analytical fluxes (black) along the grounding lines of four major ice streams draining into Filchner-Ronne Ice Shelf. All values are smoothed applying a moving average filter with span 5. Locations where the flux formula provides unphysical results are marked in grey. Plotted grounding line segments are located as displayed in the inset with western margins indicated by a yellow dot.

[revised manuscript text omitted]

---

## Author Response (AR2)

**Response to Editor Decision**
**Date: 16 August 2018**
**By R. Reese, R. Winkelmann, G. H. Gudmundsson**

Journal: TC
Title: Grounding-line flux formula applied as a flux condition in numerical simulations fails for buttressed Antarctic ice streams
Author(s): R. Reese, R. Winkelmann, G. H. Gudmundsson
MS No.: tc-2017-289
MS Type: Research article

Comments to the Author:
Dear Ronja,

Thanks for this new version of your manuscript and your reply to both reviews, which have been checked again by reviewer #1.

I think that the current version of your paper can be accepted in The Cryosphere. Nevertheless, after this last careful reading I found some typos and have one suggestion that you might want to include to improve the final version of the paper. See below.

Best regards, Olivier Gagliardini

Main suggestion: In the discussion, the sensitivity of the results are studied as a function of different parameters (sliding exponent, regularisation, mesh), but the fact that the model is not relaxed after inversion might have much more influence on your results? It is well know that flux divergence are very often not realistic from the inversion (even if the horizontal velocity are close to the observed ones, vertical ones might be far from observations at some place, inducing potentially large surface adjustment), such that a short relaxation period of the free surface is required before running any transient simulations. How much your results would be influenced by such a relaxation procedure could be discussed, at the same level than the parameter sensitivity experiments.

Typos:

- page 1, line 9: complexed-valued -> complex-valued ?

- page 2, line 31: no capital letter after ":"

- Figure 1 could be improved: reduce the space between the different panel, the size of label should be similar to the main text size and homogeneous for all panels

- page 6, line 27: can you name $\tau_f$?

- page 9, line 12: theta must be negative -> a negative value for theta is obtained (?)

- In Figs. B3 and B5, it is not clearly mentioned that it is $\theta_1$ that is plotted. Also, in the legend of Figs. B4 and B5, extended is used instead of analytical.

Dear Olivier Gagliardini,

many thanks for handling the review process of our manuscript and your helpful remarks! We are very happy that you find the current manuscript acceptable for publication in The Cryosphere after minor revisions. We addressed all issues and added figures to the supplement showing that relaxing the model does not influence our findings. Please find below the revised manuscript with all changes indicated (including the supplement).

Best regards,
Ronja Reese, Ricarda Winkelmann and Hilmar Gudmundsson

[revised manuscript text omitted]
 (left panel) and Ross Ice Shelves (right panel). In comparison to Fig. 2, the model was run forward for $1.5$ years, allowing ice thickness and grounding lines to relax. Insets indicate the ice shelves' locations in Antarctica, correspondingly. Regions where the grounding line is 'over-buttressed', that is, $\theta \leq 0$, are displayed in black. Modelled speed is plotted in gray ranging up to $1,500\ \mathrm{ma}^{-1}$. Grounding line and ice front locations are indicated in black. IS denotes ice streams, IR denotes ice rises or rumples.

[Figure]

[Figure]

**Figure S.7.** Difference between the analytical and the modeled fluxes along the grounding lines of Filchner-Ronne Ice Shelf (left panel) and Ross Ice Shelf (right panel). In comparison to Fig. 5, the model was run forward for $1.5$ years, allowing ice thickness and grounding lines to relax. Analytical fluxes are calculated based on $\theta_1$ defined in Eq. 11. In locations where the formula yields unphysical results, fluxes are set to zero. Grey arrows show the modeled ice flow. IS denotes ice streams, IR denotes ice rises or rumples. Grounding line and ice front locations are indicated in black.

---

## Author Response (AR3)

**Response to Editor Decision**
**Date: 5 September 2018**
**By R. Reese, R. Winkelmann, G. H. Gudmundsson**

Journal: TC
Title: Grounding-line flux formula applied as a flux condition in numerical simulations fails for buttressed Antarctic ice streams
Author(s): R. Reese, R. Winkelmann, G. H. Gudmundsson
MS No.: tc-2017-289
MS Type: Research article

Dear Olivier Gagliardini,

we are very happy that you find the current manuscript acceptable for publication in The Cryosphere. Please find below a latex diff showing all changes made to the manuscript. We corrected typos and specify p.4 l.18 now in the caption of Fig. S4.

We would like to thank you for editing our manuscript!

Best regards,
Ronja on behalf of the authors

[revised manuscript text omitted]

*Author contributions.* All authors designed the study. RR carried out the analysis based on an Antarctic setup and using the ice-flow model

15  Úa, both developed by GHG. RR prepared the manuscript with contributions from GHG and RW.

*Acknowledgements.* We would like to thank Christian Schoof, the anonymous reviewer and the editor Olivier Gagliardini for their helpful comments on the manuscript. 
[revised manuscript text omitted]

[Figure]

**Figure S.5.** Bivariate, normalized histogram of velocity residuals for all nodes of the Antarctic-wide mesh.

[Figure]

**Figure S.6.** Buttressing ratio $\theta_1$ along the grounding lines of Filchner-Ronne (left panel) and Ross Ice Shelves (right panel). In comparison to Fig. 2, the model was run forward for $1.5$ years, allowing ice thickness and grounding lines to relax. Insets indicate the ice shelves' locations in Antarctica, correspondingly. Regions where the grounding line is 'over-buttressed', that is, $\theta \leq 0$, are displayed in black. Modelled speed is plotted in gray ranging up to $1,500 \ \mathrm{ma}^{-1}$. Grounding line and ice front locations are indicated in black. IS denotes ice streams, IR denotes ice rises or rumples.

[Figure]

**Figure S.7.** Difference between the analytical and the modeled fluxes along the grounding lines of Filchner-Ronne Ice Shelf (left panel) and Ross Ice Shelf (right panel). In comparison to Fig. 5, the model was run forward for $1.5$ years, allowing ice thickness and grounding lines to relax. Analytical fluxes are calculated based on $\theta_1$ defined in Eq. 11. In locations where the formula yields unphysical results, fluxes are set to zero. Grey arrows show the modeled ice flow. IS denotes ice streams, IR denotes ice rises or rumples. Grounding line and ice front locations are indicated in black.